# Nuclear decoupling is part of a rapid protein-level cellular response to high-intensity mechanical loading

Hamish T.J. Gilbert [1,2], Venkatesh Mallikarjun [1,2], Oana Dobre [1,2,4], Mark R. Jackson [1,2,5], Robert Pedley [1,3], Andrew P. Gilmore [1,3], Stephen M. Richardson [2] & Joe Swift [1,2]

Studies of cellular mechano-signaling have often utilized static models that do not fully replicate the dynamics of living tissues. Here, we examine the time-dependent response of primary human mesenchymal stem cells (hMSCs) to cyclic tensile strain (CTS). At low-intensity strain (1 h, 4% CTS at 1 Hz), cell characteristics mimic responses to increased substrate stiffness. As the strain regime is intensified (frequency increased to 5 Hz), we characterize rapid establishment of a broad, structured and reversible protein-level response, even as transcription is apparently downregulated. Protein abundance is quantified coincident with changes to protein conformation and post-translational modification (PTM). Furthermore, we characterize changes to the linker of nucleoskeleton and cytoskeleton (LINC) complex that bridges the nuclear envelope, and specifically to levels and PTMs of Sad1/UNC-84 (SUN) domain-containing protein 2 (SUN2). The result of this regulation is to decouple mechano-transmission between the cytoskeleton and the nucleus, thus conferring protection to chromatin.

[1] Wellcome Centre for Cell-Matrix Research, Oxford Road, Manchester M13 9PT, UK. [2] Division of Cell Matrix Biology and Regenerative Medicine, School of Biological Sciences, Faculty of Biology, Medicine and Health, Manchester Academic Health Science Centre, University of Manchester, Manchester M13 9PL, UK. [3] Division of Molecular and Clinical Cancer Sciences, School of Medical Sciences, Faculty of Biology, Medicine and Health, Manchester Academic Health Science Centre, University of Manchester, Manchester M13 9PL, UK. [4] Present address: School of Engineering, University of Glasgow, Glasgow G12 8QQ, UK. [5] Present address: Institute of Cancer Sciences, Glasgow G61 1QH, UK. Correspondence and requests for materials should be addressed to J.S. (email: joe.swift@manchester.ac.uk)

The structures and integrity of the human body are defined by stiff tissues such as skin, muscle, cartilage and bone. Tissue mechanical properties are determined primarily by the extracellular matrix (ECM), in particular by the identities and concentrations of its constitutive proteins[1–3]. ECM properties are further modulated by protein cross-linking, post-translational modifications (PTMs), and higher-order organization. Cells resident within tissues maintain mechanical equilibrium with their environments[4,5], and the mechanical properties of cells are also regulated by the identities, concentrations, conformations and PTMs of structural intracellular proteins[1,6,7]. The characteristics of adherent cells can be influenced by physical stimulation from the surrounding ECM. Cellular protein content[1], morphology[1,8], motility[9,10] and differentiation potential[11,12] are amongst behaviors known to be affected by stiffness. Cells in living tissues experience microenvironments of diverse stiffness[5], but are also subject to deformation during activity. Cells sense and respond to mechanical signals through pathways of mechanotransduction[13,14], but must also maintain integrity and homeostasis within the tissue. A mismatch between mechanical loading and cellular regulation can contribute to pathology, such as in musculoskeletal and connective tissue disorders[15], with ageing being a significant risk factor[16].

Here, we compare responses to stiffness and mechanical loading in primary human mesenchymal stem cells (hMSCs), a cell type with important physiological and reparative roles, that have led to investigations of their therapeutic potential in tissues such as muscle[17] and heart[18]. Using mass spectrometry (MS), we identify a rapid, reversible and structured regulation of the proteome following high-intensity mechanical loading. Furthermore, we identify the Sad1/UNC-84 (SUN) domain-containing protein 2 (SUN2) as a strain-induced breakpoint in the linker of nucleoskeleton and cytoskeleton (LINC) complex of proteins that acts as a key mediator of intracellular mechano-transmission[13,19], thus enabling the nucleus to decouple from the cytoskeleton in response to intense strain.

## Results

**Strain cycle uncouples cell and nuclear morphologies.** Primary hMSCs were cultured on stiffness-controlled polyacrylamide hydrogels or silicone elastomer sheets that could be subjected to cyclic tensile strain (CTS); both substrates were collagen-I coated. hMSCs cultured for 3 days were found to spread increasingly on stiffer substrates over a physiological range (2–50 kPa; Fig. 1a, Supplementary Fig. 1a), as has been reported previously[1,20]. Cells subjected to sinusoidal, equiaxial CTS for 1 h at 1 or 2 Hz (change in strain = 4%) showed significantly increased spreading immediately after loading ($p \leq 0.05$, determined by ANOVA testing), returning to initial spread areas after 24 h (Fig. 1b, Supplementary Figs. 1b, c). Earlier reports of cell behavior following strain have described cell alignment relative to the direction of strain[21] and reorganization of focal adhesion (FA) complexes and the cytoskeleton[22,23]. As the strain applied in our system had radial symmetry, no overall alignment was observed, but increased cell spreading was consistent with previous reports describing FA activation[24]. The increase in spreading of hMSCs following dynamic straining at 1 and 2 Hz was thus similar to that observed with changes in static substrate stiffness.

To explore mechanisms that allow cells to endure more challenging mechanical environments (e.g. the mechanical environments encountered within dynamic tensile tissues, including muscle, skin and cartilage), we increased the frequency of CTS to 5 Hz (change in strain = 3.6%). The increased cell spreading observed at lower frequencies was not seen following 1 h of 5 Hz CTS (Fig. 1c). Cell spreading was significantly decreased 24 h after

treatment ($p = 0.05$, ANOVA), but cells remained attached to the substrate. Furthermore, neither cell viability nor proliferation were significantly affected (Supplementary Fig. 1d, e).

The nuclear area of hMSCs was found to increase with cell spreading on stiffer substrates (Fig. 1d). This agrees with findings reported in earlier works[1,25] and reflects the interconnected nature of the cytoskeleton and nucleoskeleton[13], which has been shown to be necessary for mechanotransduction[19]. However, we found that this correlated behavior of cell and nuclear spreading was lost in hMSCs subjected to CTS: there were no significant changes in nuclear area following 1 h of CTS at 1 or 2 Hz (Fig. 1e); and nuclear area was significantly decreased following 1 h of CTS at 5 Hz ($p = 0.003$, ANOVA; Fig. 1f, Supplementary Fig. 1f), recovering after 24 h. Under all CTS conditions, ratios of nuclear to cytoplasmic area were significantly decreased immediately following strain ($p < 0.01$, ANOVA; Fig. 1g). Thus, CTS was found to decouple the coordinated behavior of cell and nuclear spreading observed at equilibrium on stiffness-defined substrates, either through failure of the nucleus to match CTS-induced cellular spreading (CTS at 1 and 2 Hz), or through nuclear contraction while cell spreading remained constant (at 5 Hz; Fig. 1h). Dynamic loading was thus accompanied by a disruption of the mechanisms linking the cytoskeleton and nucleoskeleton.

**CTS-induced nuclear contraction requires ion channels.** Stretch activated ion-channels can enable rapid cellular responses to mechanical stimulation[26]. To investigate the role of ion channels in strain-induced nuclear contraction, we combined 5 Hz CTS with a panel of ion channel inhibitors: GdCl₃, a broad-spectrum inhibitor of stretch-activated ion channels[27]; RN9893, an inhibitor of transient receptor potential cation channel subfamily V member 4 (TRPV4)[28]; amiloride, an inhibitor of acid sensing ion channels (ASICs)[27], and GsMTx4, an inhibitor of piezo channels[29] (Supplementary Figs. 2a, b). GdCl₃, RN9893 and amiloride inhibited nuclear contraction following CTS. GsMTx4 did not prevent nuclear contraction, although earlier work has shown it to be effective in inhibiting chromatin condensation under milder loading regimes (3% uniaxial strain at 1 Hz)[29]. This suggested that activation of different ion channels may be specific to the loading regime.

CTS at 1 and 5 Hz significantly increased the texture parameter of nuclear DAPI staining (1 Hz, $p < 0.0001$ and 5 Hz, $p = 0.002$, ANOVA; Supplementary Fig. 3a–c), indicative of chromatin condensation[29,30] (comparable to the effect of divalent ions, Supplementary Fig. 3d, e). Treatment with GdCl₃ at its IC₅₀ of 10 μM[31] did not prevent changes to DAPI-stain texture following CTS at 5 Hz ($p = 0.02$, ANOVA). This contrasted with earlier characterizations of milder loading regimes, where GdCl₃ was found to block chromatin condensation, although the drug concentration was higher in this case[29]. Our finding indicated the robustness of the chromatin condensation response in cells subjected to high-intensity CTS, but also suggested that chromatin condensation and contraction of nuclear area could be caused by different mechanisms.

**Cellular responses to CTS are driven at the protein level.** In order to examine the cellular consequences of high-intensity strain, we applied -omics analyses to hMSCs following 1 h of 5 Hz CTS. Surprisingly, we found few changes to the transcriptome as assessed by RNA-Seq immediately after the treatment (Gaussian width = 0.21; Fig. 2a, Supplementary Data 1 and 2). Furthermore, gene ontology (GO) term analysis[32,33] of the affected genes suggested a general suppression of transcription and metabolism (Fig. 2b). Downregulation of transcriptional activity is consistent with our observations of chromatin condensation, and previous reports of histone-methylation mediated gene silencing in

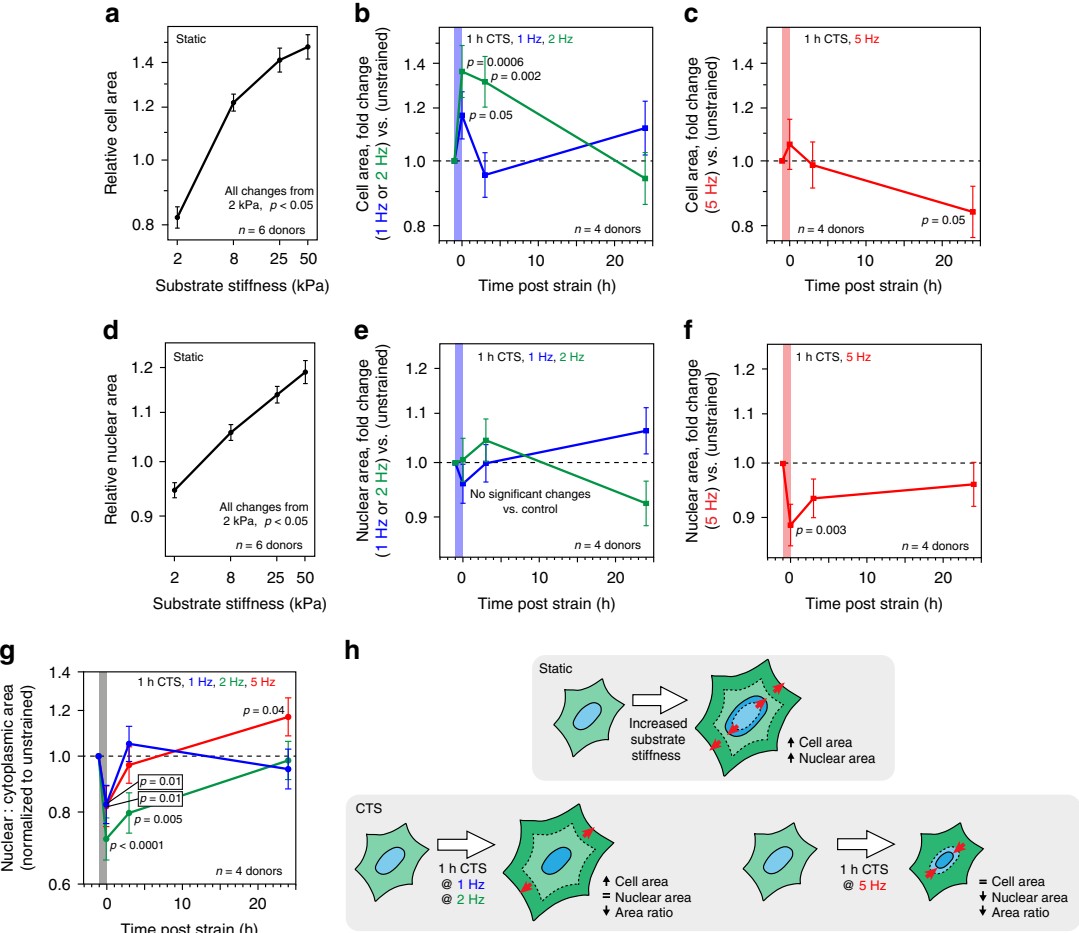

**Fig. 1** Coupled cell and nuclear morphologies are lost following cyclic tensile strain (CTS). **a** Relative areas of primary human mesenchymal stem cells (hMSCs) cultured on static substrates (collagen-I coated polyacrylamide hydrogels; 2–50 kPa; $n = 6$ donors), showing increase with substrate stiffness ($p < 0.05$). **b** Cell areas of hMSCs following low-intensity CTS (0–4% strain at 1 or 2 Hz for 1 h; $n = 4$ donors). Areas increased immediately following strain (1 Hz, $p = 0.05$; 2 Hz, $p = 0.0006$ and $p = 0.002$ at 3 h). **c** Cell areas of hMSCs following high-intensity CTS (2.6–6.2% strain at 5 Hz for 1 h; $n = 4$ donors), showing decreased 24 h post strain ($p = 0.05$). **d** Relative nuclear areas of hMSCs cultured on static substrates (collagen-I coated PA hydrogels; 2–50 kPa; $n = 6$ donors). Nuclear areas increased with substrate stiffness ($p < 0.05$). **e** Nuclear areas of hMSCs after low-intensity CTS (0–4% strain at 1 or 2 Hz for 1 h; $n = 4$ donors). **f** Nuclear areas of hMSCs cultured following high-intensity CTS (2.6–6.2% strain at 5 Hz for 1 h; $n = 4$ donors). Nuclear areas decreased immediately following strain treatment ($p = 0.003$). **g** Nuclear to cytoplasmic area ratio of hMSCs following CTS (1 h; 0–4% strain at 1 or 2 Hz or 2.6–6.2% strain at 5 Hz; $n = 4$ donors). Area ratios decreased immediately following strain treatment for all CTS frequencies (1 Hz, $p = 0.01$; 2 Hz, $p < 0.0001$; 5 Hz, $p = 0.01$). Only the 5 Hz treatment group showed an increased ratio at 24 h (5 Hz, $p = 0.04$). All CTS experiments normalized to unstrained controls; data displayed as mean ± s.e.m.; $p$-values determined from ANOVA. See Supplementary Fig. 1a–c, f for example cell images and donor-to-donor variability; Supplementary Table 1 for sample sizes. **h** In summary, cell and nuclear areas appeared coupled on increasingly stiff substrates. Low-intensity CTS increased cell area, but did not change nuclear area; high-intensity CTS caused nuclear area to decrease independently of cell area

endothelial progenitor cells subjected to low frequency (0.1 Hz) strain cycling[34]. The distribution of changes to gene expression was narrowed 24 h after CTS (Gaussian width = 0.14; Supplementary Fig. 4a).

In contrast to the analysis of transcript levels, analysis of the intracellular proteome by MS, showed greater changes relative to unstrained controls (Fig. 2c, Supplementary Fig. 4b, Supplementary Data 3 and 4). A Gaussian fit to the distribution of protein fold changes had a width of 0.61, indicating a greater perturbation to proteome than transcriptome. Analysis of the Reactome pathways significantly affected by CTS (pathways with over-representation of affected proteins; Bayes-modified $t$-tests with Benjamini–Hochberg false discovery rate (BH-FDR)-corrected $p < 0.05$)[35,36] showed that ontologies relating to metabolism of both protein and RNA, signal transduction, and response to external stimuli were downregulated (Fig. 2d). Changes to the transcriptome and proteome following CTS were not correlated (R-squared = 0.002; Fig. 2e), indicating a

post-transcriptional regulation of protein levels. The proteome returned towards the control state after 24 h (Gaussian width = 0.25; Supplementary Figs. 4c–e).

The time-resolved proteomic response to 5 Hz CTS was further classified by K-means clustering (Fig. 2f, Supplementary Fig. 4f). Clusters of protein levels were identified with: (cluster 1) an immediate but unsustained decrease, enriched for Reactome annotations associated with translation, protein folding, and mechanisms of actin and tubulin folding; (cluster 2) an initial but unsustained increase, enriched for an annotation of metallothionein binding (associated with the management of oxidative stress[37]); and (cluster 3) an immediate decrease and slow recovery, with enrichment of annotations for translation and regulation of the Slit/Robo signaling pathway (associated with cell polarity and cytoskeletal dynamics[38]). Taken as a whole, this analysis shows a complex, time-resolved, and structured protein-level response to cellular stress management.

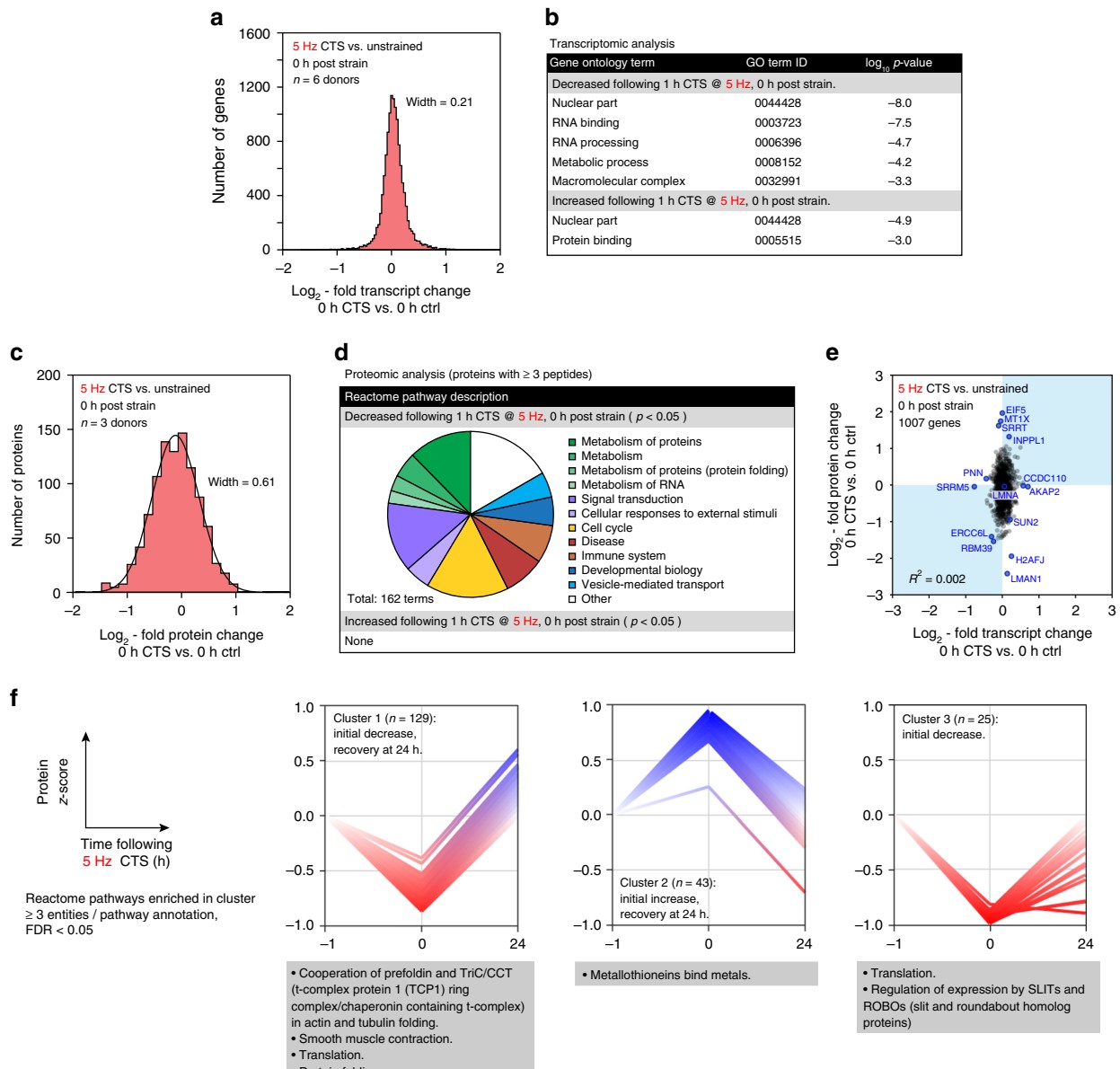

**Fig. 2** Changes to the proteome following CTS are uncorrelated to transcript. **a** Histogram of changes to mRNA in primary hMSCs immediately following high-intensity CTS (1 h at 5 Hz, 2.6–6.2 % strain; $n = 6$ donors); data displayed as log$_2$-fold transcript change CTS versus unstrained controls. The width of a Gaussian curve fitted to the distribution was 0.21. There were no significant changes in *SUN2*. See Supplementary Data 1 and 2. **b** Gene ontology (GO) term analysis of the transcripts significantly affected by CTS. Annotations suggest nuclear remodeling (nuclear component terms are both up- and down-regulated), and down-regulation of RNA processing and metabolism. **c** Histogram of proteomic changes quantified by mass spectrometry (MS) immediately following CTS (1 h at 5 Hz, 2.6–6.2 % strain; $n = 3$ donors). Data displayed as log$_2$-fold change following CTS, versus unstrained controls. Proteins quantified by ≥3 peptides. See Supplementary Fig. 4b for volcano plot and Supplementary Data 3 and 4. A Gaussian curve fit to the distribution had a width of 0.61 and maximum at $-0.11 \pm 0.01$. **d** Analysis of Reactome pathways significantly affected at the protein level following CTS ($p < 0.05$; proteins quantified by ≥3 peptides; $n = 3$ donors). Proteins associated with cellular metabolic processes were decreased following strain, including protein and RNA associated metabolism. **e** Correlation plot between proteome and transcriptome immediately following CTS (1007 genes quantified by RNA-seq and in proteomics by ≥3 peptides; selected outlying genes/proteins of interest are annotated; R-squared = 0.002). The distribution of changes to protein levels was broader than, and uncorrelated with, transcript changes. Transcript and protein levels were partially recovered 24 h after CTS (Supplementary Fig. 4a, c–e). **f** K-means clustering was used to group quantified proteins based on their response to CTS after 0 and 24 h. Analysis showed four possible clusters to be most appropriate for this dataset (Supplementary Fig. 4f). Three clusters are shown annotated with significant Reactome enrichments; ≥3 proteins per annotation; false discovery rate (FDR) <0.05. $p$-values were calculated using empirical Bayes-modified $t$-tests with Benjamini–Hochberg correction

For comparison, we also examined proteomic changes in response to low-intensity CTS (1 h at 1 Hz, change in strain = 4.0%). We found changes to 1 Hz CTS to be less pronounced than those induced at 5 Hz (Gaussian width = 0.30; Supplementary Figs. 5a, b, Supplementary Data 8 and 9), and although Reactome analysis showed similar pathways to be affected (compare Fig. 2d to Supplementary Fig. 5c), we noted that many of the significantly affected proteins were associated with the cytoskeleton. CTS at 1 Hz caused a significant increase in levels of actin (ACTB), vimentin (VIM), tubulin alpha-1B chain (TUBA1B) (all $p <$

0.0001, Bayes-modified $t$-test, BH-FDR correction), dynamin-2 (DNM2, $p = 0.04$, Bayes-modified $t$-test, BH-FDR correction) and the nucleoskeletal protein lamin-A/C (henceforth the abbreviation LMNA will be used to refer to the total protein products of the *LMNA* gene, comprising of both lamin A and C spliceforms; $p = 0.008$, Bayes-modified $t$-test, BH-FDR correction); myosin light chain 6B (MYL6B) and the mechano-responsive transcriptional coactivator YAP1 were downregulated ($p = 0.001$ and 0.03, respectively, Bayes-modified $t$-tests, BH-FDR correction). Where these proteins were also detected in the 5 Hz experiment, only VIM was significantly affected (down-regulated, $p = 0.002$, Bayes-modified $t$-test, BH-FDR correction). As was observed following 5 Hz CTS, the proteome partially recovered 24 h following strain at 1 Hz (Supplementary Fig. 5d, e; Gaussian width = 0.26), although some changes to cytoskeletal proteins persisted (TUBA1B remained elevated, ACTB was decreased; both $p < 0.0001$, Bayes-modified $t$-tests, BH-FDR correction). Transcripts associated with the cytoskeleton were also affected: vimentin (*VIM*) was increased immediately following CTS at 1 Hz ($p = 0.0003$, ANOVA) and alpha-actin-2 (*ACTA2*) was increased after 24 h ($p = 0.004$, AVOVA) (Supplementary Fig. 5f, g). These results are consistent with previously observed changes to cell morphology, and earlier characterizations of cellular responses to strain[23,39] and substrate stiffness[1], which were proposed to increase nucleoskeletal and cytoskeletal robustness to stress.

**CTS at 5 Hz causes changes to protein conformation and PTMs**. As changes in protein conformation are important to mechanotransduction[14], MS was performed following protein labeling with monobromobimane (mBBr), which by selectively labeling solvent-exposed cysteine residues, acts as an indicator of protein folding (Fig. 3a). MS was used to both identify mBBr-labeled proteins and quantify differential labeling in hMSCs following CTS at 5 Hz, relative to unstrained controls (Fig. 3b, Supplementary Data 5). The histogram of $\log_2$-fold changes in mBBr labeling showed a broad distribution of CTS-induced changes to mBBr reactivity, with labeling increased on average immediately following strain. The distribution was narrowed and centered at about zero 24 h after CTS, indicating a recovery of protein folding. Earlier applications of mBBr labeling have been used to identify force-dependent unfolding of domains in spectrin[40] and nuclear LMNA[1]. Labeling of Cys-522 in the Ig-folded domain of LMNA was previously used to report on the deformation of isolated nuclei subjected to shear stress in a rheometer[1]. We found the labeling of Cys-522 to be increased 1.1-fold immediately following strain ($p < 0.001$, Bayes-modified $t$-test, BH-FDR correction). We correlated changes to mBBr-labeled cysteine site occupancy and changes to total quantities of the parent proteins (Fig. 3c). This analysis showed that despite the suggestion of a link between CTS-induced changes to protein conformation and stability (rates of translation vs. turnover) on average (Figs. 2c and 3b), correlation on a protein-by-protein basis was low (Fig. 3c).

Changes to endogenous PTMs were quantified by MS in the same experiment. A histogram of $\log_2$-fold changes to phospho-site occupancy following CTS at 5 Hz versus unstrained controls (Fig. 3d, Supplementary Data 6) showed increased phosphorylation. Phosphorylation of LMNA has been shown to be lower on stiffer substrates where total LMNA was increased[1,7]. However, here we detected a modest (~1.1-fold) but significant increase in phosphorylation at S22, S390, S392, and S636, and no change in LMNA abundance. A plot of all changes in phosphosite occupancy vs. changes in abundance of the phosphorylated protein (Fig. 3e) did not exhibit general correlation, indicating

that although phosphorylation may in many cases be mechano-sensitive, it does not necessarily regulate turnover. A similar analysis of oxidized peptides showed that oxidation was increased immediately following CTS (Fig. 3f, Supplementary Data 7), and in a third of detected sites of oxidation, increased oxidation correlated with decreased protein levels (points in the top left quadrant of Fig. 3g). Previous work has established that cyclic stretching can increase levels of reactive oxygen species (ROS) in a range of cell types through activation of nicotinamide adenine dinucleotide phosphate (NADPH) oxidase systems and mito-chondria[41]. ROS have an important role in signal transduction, for example during vascularization, but can contribute to oxidative damage to lipids, proteins, and DNA. This potential for oxidative damage is perhaps consistent with the upregulation of protective metallothioneins (Fig. 2f). Profiles of both phosphorylation and oxidation became similar to controls 24 h after CTS (Fig. 3d, f).

**CTS at 5 Hz disrupts the LINC complex**. Having detected systematic responses to CTS, we sought to identify specific cases where protein regulation modulated mechano-transmission to the nucleus. There is a continuous system of structural proteins that run from the ECM through to chromatin[13,19]. A central feature of this pathway is the LINC complex, which spans the nuclear envelope (NE) and includes the outer nuclear membrane (ONM) nesprin proteins, which bind to cytoskeletal components in the cytoplasm and have Klarsicht, ANC-1, and Syne homology (KASH) domains that extend into the perinuclear space of the nuclear envelope[42]. The KASH domains bind to the SUN domains of the inner nuclear membrane (INM) SUN-domain containing proteins, which in turn bind to the nuclear lamina within the nucleoplasm[43]. The nuclear lamina is composed of the intermediate filament lamin proteins that confer structural integrity to the nucleus[2,44,45], and also interface with chromatin and a range of regulatory and NE associated proteins[1,46]. The complete system of protein linkages enables nuclear positioning[47] and acts as a conduit for mechanical signals to regulate the genome[19,46].

LINC and NE protein levels were quantified by MS following 1 h of CTS at 5 Hz, relative to unstrained controls (Fig. 4a). The levels of the SUN protein SUN2 was reduced to 52% of control levels ($p < 0.0001$, Bayes-modified $t$-test, BH-FDR correction). Note that SUN2 was not affected by 1 Hz CTS (Supplementary Fig. 5a, d). We also quantified the levels of proteins located specifically at the NE using immunofluorescence (IF) imaging (Fig. 4b, Supplementary Fig. 6a–e). IF confirmed that SUN2 levels were decreased at the NE following 1 h of CTS ($p = 0.03$, ANOVA). Emerin (EMD), which has a role in the mechanical stimulation of the serum response factor (SRF) pathway[48], was significantly enriched at the NE ($p < 0.0001$, ANOVA), consistent with previous reports[34].

**Mechano-sensitive phosphosites in SUN2 lamin-binding domain**. To understand the regulation of SUN2 and its role in the nuclear decoupling phenomena, we examined the response of hMSCs to shorter durations of CTS at 5 Hz. IF showed that SUN2 was significantly reduced at the NE after 1 min of CTS ($p = 0.002$, ANOVA; Fig. 4c). This preceded changes to the ratio of nuclear to cytoplasmic area, which were not significantly reduced within 10 min of CTS (Fig. 4d). PTMs have been shown to regulate the assembly of nuclear proteins such as LMNA[1,7], so we used MS to search for modifications to SUN2 following 1 h of CTS at 5 Hz (coverage of the SUN2 amino acid sequence shown in Fig. 5a, b). This analysis uncovered four strain-responsive phosphorylation sites within the lamin-binding domain of SUN2; modifications

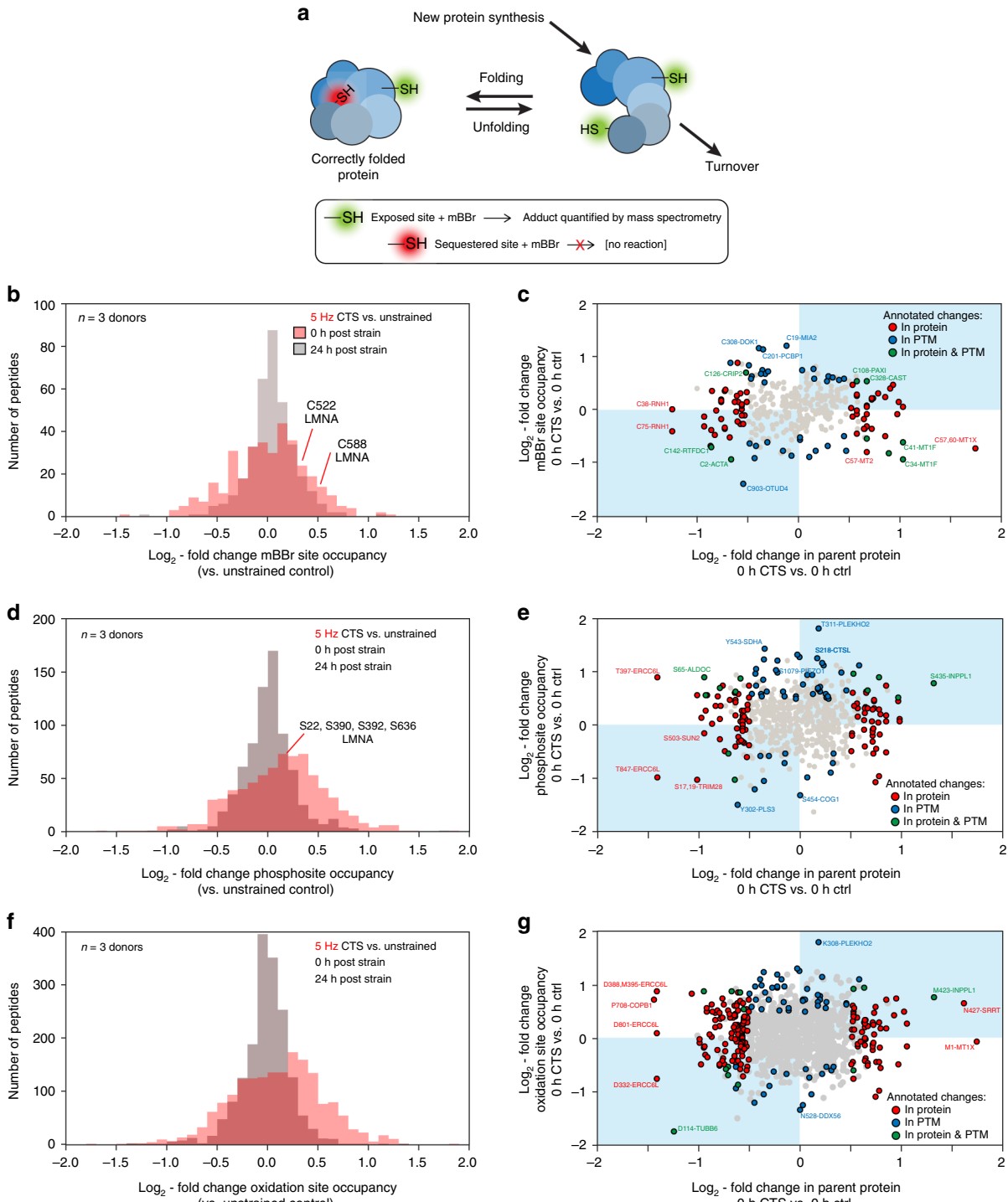

were found that occurred immediately (decrease in pS12, $p = 0.007$; increase in pS21, $p < 0.0001$; and increase in pS38, $p < 0.0001$) and some persisted 24 h following CTS (increase in pT9, $p = 0.003$; decrease in pS12, $p < 0.0001$; and increase in pS21, $p < 0.0001$; $p$-values derived from Bayes-modified $t$-tests with BH-FDR-correction; Fig. 5c).

Taken together, this evidence suggests a putative mechanism (Fig. 5d) whereby high-intensity CTS causes rapid loss of SUN2 from the NE – potentially mediated by phosphorylation of the lamin-binding domain – followed by a slower remodeling of cell and nuclear morphology and the cellular proteome (including turnover of SUN2). The composition of the nuclear lamina has

been used previously as a readout of nuclear adaption to the mechanical properties of the cellular microenvironment, with an increased ratio of A-type to B-type lamins being indicative of nuclear stiffening in response to stiffness[1,44]. We used IF to quantify the ratio of total LMNA to lamin-B1 after 1 h of CTS at 5 Hz, finding it to be significantly decreased ($p = 0.007$, ANOVA; Supplementary Fig. 7a). Our findings contrast earlier reports[1] of responses within the nuclear lamina to increasing substrate stiffness (summarized in Supplementary Fig. 7b) and suggest that SUN2-mediated nuclear decoupling could desensitize the lamina to mechanical stimulation. Lastly in our investigations into causation, we found that treatment with GdCl3 – shown to

**Fig. 3** Post-translational modification (PTM) states respond to CTS. **a** Schematic diagram of differential labeling of sulfhydryl groups by monobromobimane (mBBr)[1, 40]. Exposed cysteine residues are rapidly labeled, but those sequestered within folded proteins are unreactive; changes to the extent of labeling are indicative of altered protein conformation in response to a stress condition. Changes in protein quantity are determined by rates of synthesis versus turnover. **b** Histogram showing $\log_2$-fold changes to mBBr labeling site occupancy immediately and 24 h after hMSCs were subjected to CTS (1 h at 5 Hz, 2.6–6.2 % strain; $n = 3$ donors), relative to unstrained controls. The distribution immediately following CTS was broad and displaced to the right. After 24 h, the distribution was narrowed and centered around zero. Annotations indicate two sites within LMNA with significantly increased modification ($p <$ 0.05). **c** Correlation between changes to mBBr labeling site occupancy and the quantity of the parent protein (i.e. source of the labeled peptide), immediately following CTS. **d** Histogram showing changes to phosphosite occupancy immediately and 24 h after CTS (1 h at 5 Hz, 2.6–6.2% strain; $n = 3$ donors), relative to unstrained controls. All phosphorylation sites shown have been curated previously in the PhosphoSitePlus database[70]. The distribution was shifted to the right immediately following CTS. **e** Correlation between phosphorylation and protein quantity. **f** Histogram showing changes to oxidation-site occupancy immediately and 24 h after CTS (1 h at 5 Hz, 2.6–6.2 % strain; $n = 3$ donors), relative to unstrained controls. **g** Correlation between oxidation and protein quantity; in 33% of cases, increased oxidation correlated with decreased protein levels (points in the top left quadrant). In **c**, **e**, and **g**, data points are annotated as indicated in the legend if |$\log_2$-fold change| > 0.5 and $p < 0.05$, otherwise they are shown in gray; labels indicate modified sites within proteins. $p$-values were calculated using empirical Bayes-modified $t$-tests with Benjamini–Hochberg correction. See Supplementary Data 3–7

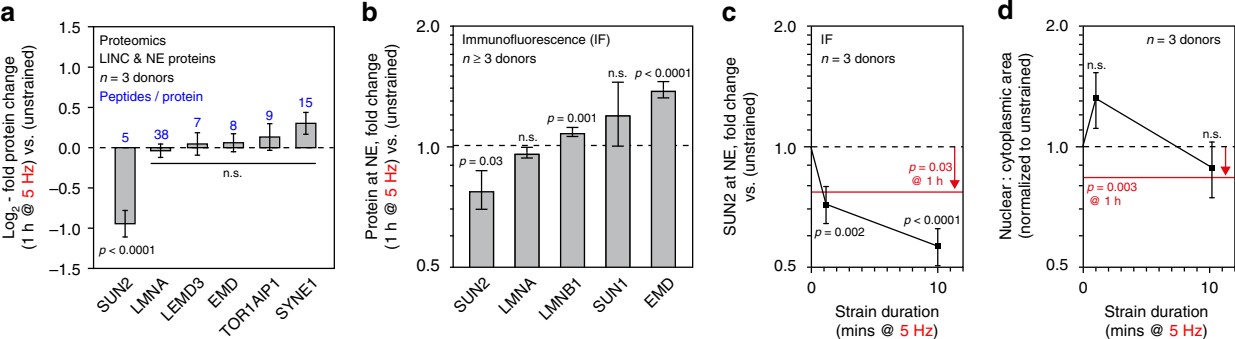

**Fig. 4** CTS causes loss from the nuclear envelope (NE) and turnover of SUN2 protein. **a** Changes to linker of nucleoskeleton and cytoskeleton (LINC) complex and NE proteins in primary hMSCs, detected and quantified by MS immediately following CTS (1 h at 5 Hz, 2.6–6.2% strain; $n = 3$ donors), relative to unstrained controls. SUN2 was found to be significantly down-regulated, but was recovered 24 h after CTS (Supplemental Fig. 3d). Numbers in blue indicate the number of peptides detected per protein identity. $p$-values were calculated using empirical Bayes-modified t-tests with Benjamini–Hochberg correction. See Supplementary Data 4. **b** Immunofluorescence (IF) quantification of proteins localized at the NE immediately following CTS (1 h at 5 Hz, 2.6–6.2% strain; $n = 3$ donors for SUN1, SUN2, LMNA, and LMNB1; $n = 4$ donors for EMD; see Supplementary Figs. 6a–e for data distributions and donor-to-donor variation). **c** IF quantification of SUN2 at the NE following 1 and 10 min of CTS at 5 Hz (2.6–6.2% strain; $n = 3$ donors). Red line indicates SUN2 levels following 1 h of CTS ($p = 0.03$). Significant loss of SUN2 at the NE occurred within 1 min (1 min, $p = 0.002$; 10 min, $p < 0.0001$). **d** Nuclear to cytoplasmic area ratios quantified following 1 and 10 min of 5 Hz CTS. Red line indicates area ratio following 1 h of CTS ($p = 0.003$; Fig. 1f). These results indicate that loss of SUN2 from the NE precedes changes to cellular morphology. $p$-values in **b**–**d** determined from linear models (ANOVA tests). All plots show mean ± s.e.m.; see Supplementary Table 1 for sample sizes

prevent contraction of nuclear area following CTS (Supplementary Fig. 2a, b) – also prevented the loss of SUN2 (Supplementary Fig. 8a, b).

**SUN2 acts upstream of chromatin and cytoskeletal regulation**. As the level of SUN2 was decreased in response to CTS, we sought to investigate how this could affect cellular function. We quantified the effects of siRNA-mediated knockdown (KD) of SUN2 on primary hMSCs by MS, comparing two siRNAs to increase confidence of identifying on-target effects (Fig. 6a, Supplementary Data 10 and 11). The siRNAs depleted SUN2 to 35% of scrambled controls (Supplementary Fig. 9a, b). A Reactome pathway analysis of both KDs (Fig. 6b) identified significant perturbations to processes with the following annotations: polycomb repressive complex 2 (PRC2) methylates histones and DNA; protein lysine methyltransferases (PKMTs) methylate histone lysines; and, Rho GTPases activate IQ motif containing GTPase activating proteins (IQGAPs) (all $p$-values < 0.05, Bayes-modified $t$-tests, BH-FDR correction). These annotations suggested that SUN2 could be upstream of chromatin regulation and nucleus-to-cytoskeleton (inside-to-outside) signaling, consistent with our earlier observations. A scatter plot of protein fold changes following SUN2 KD versus CTS showed correlation in

cytoskeletal proteins (Fig. 6c), suggesting SUN2 regulation following CTS could be upstream of cytoskeletal remodeling.

We also investigated the effects of SUN2 overexpression (OE), using immortalized hMSCs (Y201 line) that maintain the multipotency and mechano-responsiveness of primary MSCs[49,50]. The proteomes of Y201 cells were quantified with doxycycline-induced OE of SUN2 to 160% and 410% of control levels (Fig. 6d, Supplementary Fig. 9c, d, Supplementary Data 12 and 13). While the SUN2 KD caused an increase in a number of cytoskeletal proteins, SUN2 OE caused a corresponding decrease in many of the same proteins (including filamin-A, plectin and vimentin), suggestive of a compensatory mechanism within the cytoskeleton. Reactome analysis following SUN2 OE was less specific in its effects than the KD, but interestingly suggested an up-regulation of pathways associated with DNA repair (Fig. 6e). MS also revealed changes to the phosphorylation state of SUN2 following OE (Fig. 6f): consistent with OE driving removal of excess protein, we found the same sites to be affected as when SUN2 was lost following CTS (Fig. 5c).

**SUN2 modulates nuclear mechano-transmission and DNA damage**. To investigate the role of SUN2 in mechano-transmission to the nucleus, we examined the relationship

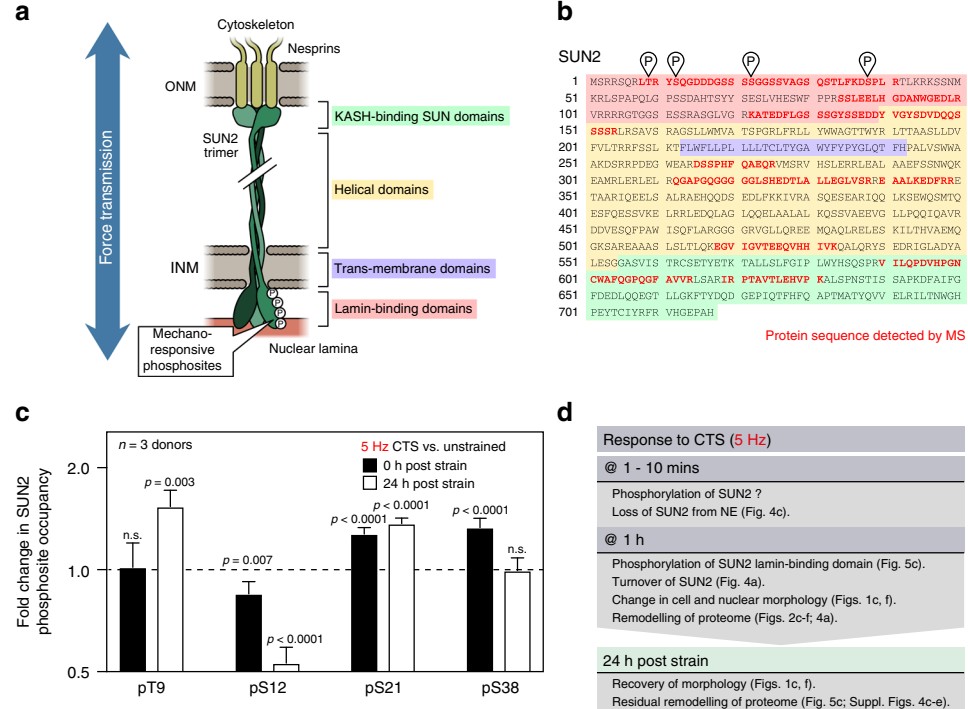

**Fig. 5** The lamin-binding domain of SUN2 contains mechano-sensitive phosphorylation sites. **a** Schematic of SUN2 protein trimer spanning the NE and linking between the nuclear lamina and nesprins[71]. MS detected four mechano-responsive phosphorylation sites within the lamin-binding domain of SUN2. ONM = outer nuclear membrane; INM = inner nuclear membrane. **b** Amino acid sequence of SUN2 protein. Red indicates regions of the sequence detected by MS, showing coverage of domains across the length of the protein; phosphosites are also indicated. Domains are indicated according to the scheme shown in figure part (**a**). **c** Quantification of phosphorylation at sites in SUN2 (pThr9, pSer12, pSer21, pSer38) immediately, and 24 h following, CTS (1 h at 5 Hz, 2.6–6.2% strain; $n = 3$ donors). pS12, pS21 and pS38 showed an immediate response to strain ($p = 0.007$, <0.0001, and <0.0001, respectively); changes to pT9, pS12, and pS21 suggested more persistent remodeling ($p = 0.003$, <0.0001, and <0.0001, respectively). Data displayed as mean and s.e.m.; $p$-values were calculated using empirical Bayes-modified $t$-tests with Benjamini–Hochberg correction. **d** Summary of evidence that suggests a sequential mechanism for nuclear decoupling, and subsequent recovery, following high-intensity CTS

between SUN2 levels and cellular morphology. Y201 cells were cultured on plastic for three days following KD, OE and rescue of SUN2 levels, and compared to controls (Fig. 7a); modulation of SUN2 level at the NE was confirmed in all cases by IF (Supplementary Fig. 10a, b). Previous reports have linked SUN2 OE with abnormally shaped nuclei[51] and we found a significant reduction in nuclear form factor in cells subjected to SUN2 KD ($p = 0.002$, ANOVA followed by Dunnett's multiple comparison tests; Supplementary Fig. 10c). Consistent with the reduction in nuclear size following SUN2 depletion after CTS at 5 Hz, we found a weak positive scaling relationship between SUN2 level and nuclear area (Fig. 7b and Supplementary Fig. 10d). However, the effect of SUN2 level on cytoplasmic area was stronger, in keeping with our observations of SUN2-induced inside-to-outside remodeling, and dominated the scaling of the nuclear to cytoplasmic ratio (Fig. 7c, d, Supplementary Fig. 10d, e). This contrast between the change in nuclear to cytoplasmic area ratio following CTS (Fig. 1g) versus remodeling following imposed modulation of SUN2 levels (Fig. 7d) perhaps reflects the difference in time scales over which these processes occur. Consistent with an interpretation of A-type lamins as reporters of a functioning mechanical linkage between the cytoskeleton and the nucleus[1], SUN2 OE led to the loss of LMNA at the NE (Supplementary Fig. 10f–h).

Finally, we sought to determine how perturbation of SUN2 could affect cellular responses to CTS. We found that SUN2 KD in primary hMSCs was sufficient to prevent changes to the nuclear to cytoplasmic area ratio following 1 h of CTS at 5 Hz (Fig. 8a, b, Supplementary Fig. 11a–e). SUN2-depletion significantly decreased strain-induced changes to nuclear texture,

indicating reduced chromatin condensation ($p < 0.0001$ for both siRNAs, ANOVA; Fig. 8c, Supplementary Fig. 11f). Likewise, SUN2 OE in immortalised hMSCs blocked the changes to the nuclear to cytoplasmic area ratio observed in controls cells ($p = 0.03$, ANOVA and Dunnett's multiple comparison tests) following 1 h of CTS at 5 Hz (with recovery after 24 h, Fig. 8d, e, Supplementary Fig. 11g–i). Rescue of SUN2 expression levels restored the capacity to decouple nuclei from the cytoskeleton following CTS ($p < 0.0001$, ANOVA and Dunnett's multiple comparison tests; Fig. 8d, e), confirming the importance of correct SUN2 expression levels for this phenomenon to occur. SUN2 OE was also found to prevent the increase to nuclear texture, associated with chromatin condensation, that was caused by CTS ($p = 0.01$, ANOVA and Dunnett's multiple comparison tests; Fig. 8f).

Mechanical strain has been shown to cause DNA damage, inducing apoptosis in vascular smooth muscle cells[52], and causing the accumulation of damage to DNA and chromatin in nuclei subjected to extreme deformation as cells migrate through constricted environments[53–55]. We were surprised, therefore, to find that CTS here resulted in a small but significant decrease in the intensity of γH2AX staining in primary ($p = 0.03$, ANOVA) and immortalised MSCs ($p = 0.0002$, ANOVA), suggestive of a protective effect (Fig. 9a, b, Supplementary Fig. 12a). However, we found that the OE of SUN2 in immortalised hMSCs shown to override the decoupling response to CTS raised the baseline level of γH2AX staining ($p < 0.0001$, ANOVA and Dunnett's multiple comparison tests) and caused staining to be further increased immediately following CTS ($p < 0.0001$, ANOVA and Dunnett's

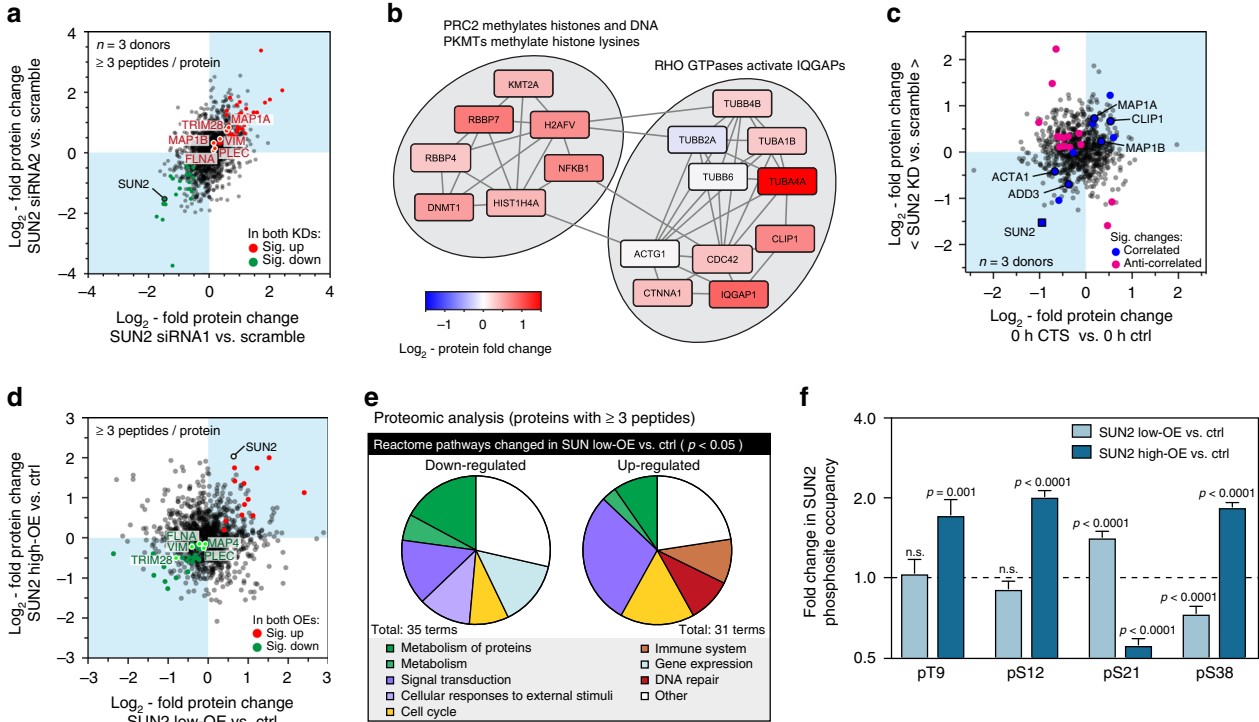

**Fig. 6** Knockdown (KD) and overexpression (OE) of SUN2 affect the cellular proteome. **a** MS characterization of SUN2 KD on protein levels in primary hMSCs. Plot shows correlation between log$_2$-fold changes to the proteome following SUN2 KD with two siRNAs, siRNA1 and siRNA2, each measured relative to scrambled controls ($n = 3$ donors; see Supplementary Figs. 9a, b for histograms). Data points annotated as indicated in the legend where significant changes occurred ($p < 0.05$), otherwise they are shown in gray. **b** Pathways identified in the Reactome database[35] as significantly affected by both SUN2 KDs, relative to scrambled controls, shown with fold changes to constitutive proteins. **c** Correlation between changes to protein levels following SUN2 KD (mean of both siRNA treatments, relative to scrambled controls) and immediately following CTS (1 h, 2.6–6.2% strain at 5 Hz relative to unstrained controls). Correlated and anti-correlated data annotated as indicated in the legend where significant changes occurred ($p < 0.05$; $n = 3$ donors), otherwise they are shown in gray. **d** MS characterization of SUN2 OE in an immortalized human MSC line. Plot shows correlation between log$_2$-fold changes to the proteome following SUN2 OE with two separate expression levels, low (160% of control following 3 days of DOX treatment) and high (410% of control following 4 days of DOX treatment), measured relative to vehicle-only controls ($n = 3$ replicates; see Supplementary Figs. 9c, d for histograms). Data points annotated as indicated in the legend where significant changes occurred ($p < 0.05$), otherwise they are shown in gray. **e** Analysis of Reactome pathways significantly affected at the protein level following SUN2 low-OE ($p < 0.05$). Protein quantification in **a**–**e** based on ≥3 peptides-per-protein. **f** Quantification of SUN2 phosphosites following low and high OE, as characterized in **d**. pS21 and pS38 were significantly affected by low SUN2 OE ($p < 0.0001$); high OE significantly affected pT9 ($p = 0.001$), pS12, sS21, and pS38 ($p < 0.0001$). $n = 3$ technical replicates. Data displayed as mean and s.e.m. All $p$-values were calculated using empirical Bayes-modified $t$-tests with Benjamini–Hochberg correction. See Supplementary Data 10–13

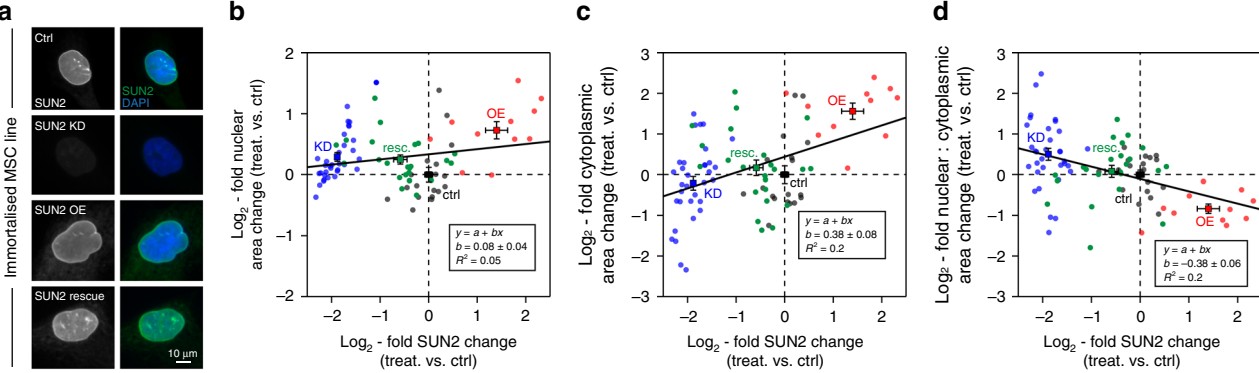

**Fig. 7** Cell and nuclear areas are both affected by perturbation to SUN2 levels. **a** Representative images of immortalized MSC nuclei stained against DAPI and SUN2 following control, SUN2 KD (siRNA2), OE and rescue (OE + siRNA2) treatments. Scatter plots showing log$_2$-fold change to **b** nuclear area, **c** cytoplasmic area, and **d** nuclear to cytoplasmic area ratio vs. log$_2$-fold change to SUN2 expression, in immortalized MSC nuclei (control, SUN2 KD, OE, and rescue), relative to control levels. Data points indicate individual nuclei (≥11 analysed per condition); annotated points show mean ± s.e.m. for each condition; lines show linear regression analysis (with annotation of gradient, $b$ ± standard deviation). See Supplementary Fig. 10b–e for data distributions and statistics. Nuclear area showed weak positive power law scaling with SUN2 expression ($b = 0.08 ± 0.04$). However, as the cytoplasmic area showed stronger scaling with SUN2 expression ($b = 0.38 ± 0.08$), the nuclear to cytoplasmic area ratio had overall negative scaling ($b = -0.38 ± 0.06$). These results suggest a pronounced inside-to-outside influence of SUN2 levels on the cytoskeleton

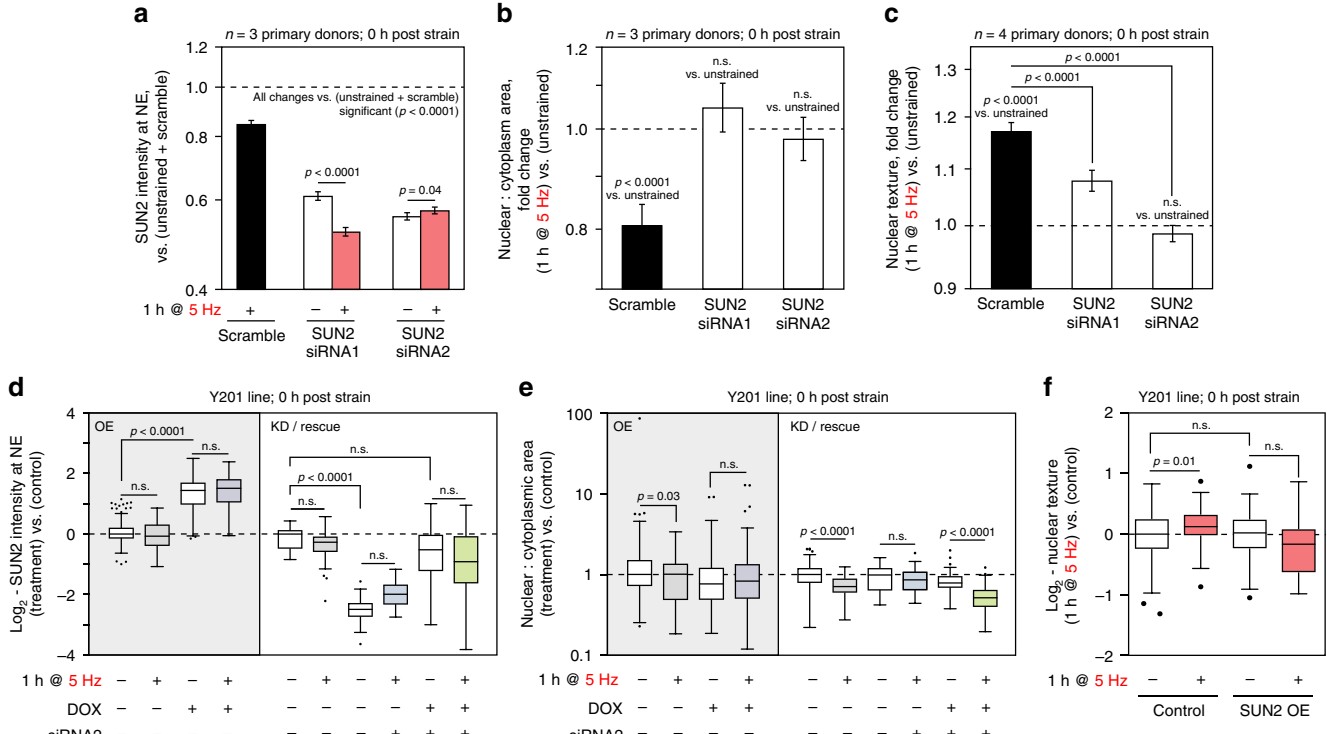

**Fig. 8** Responses to CTS are blocked by KD or OE of SUN2. **a** SUN2 levels quantified by IF at the NE in primary hMSCs, following SUN2 KD (comparing two siRNA sequences) and in combination with CTS (1 h, 2.6–6.2% strain at 5 Hz; $n = 3$ donors). Both siRNAs were effective against SUN2 ($p < 0.0001$). SUN2 level in the less potent KD (siRNA1) was further decreased by CTS ($p < 0.0001$); the more efficient KD (siRNA2) showed a small increase ($p = 0.04$). **b** Ratios of nuclear to cytoplasmic areas in primary hMSCs following SUN2 KD and CTS ($n = 3$ donors). SUN2 KD blocked the CTS-induced decrease to ratio ($p < 0.0001$ for scrambled vs. CTS). **c** Changes to nuclear texture in primary hMSCs following SUN2 KD and CTS ($n = 4$ donors). SUN2 KD significantly reduced strain-induced changes in nuclear texture ($p < 0.0001$ for SUN2 KD vs. scrambled and CTS). Plots show mean ± s.e.m.; $p$-values determined from linear modeling (ANOVA). See Supplementary Figs. 11a–f for data distributions and variation between donors. **d** Quantification of SUN2 at the NE in immortalised hMSCs with inducible SUN2 OE ($p < 0.0001$), SUN2 KD (siRNA2) ($p < 0.0001$) and induced rescue of SUN2 KD, immediately following 5 Hz CTS. **e** Ratios of nuclear to cytoplasmic areas in immortalised hMSCs with SUN2 OE, SUN2 KD (siRNA2) and SUN2 rescue, immediately following CTS. Ratios were significantly decreased in control cells following CTS ($p = 0.03$ and $p < 0.0001$), but not after OE or KD of SUN2. Rescue of SUN2 levels restored the response ($p < 0.0001$) (see Supplementary Figs. 11g–i). **d** and **e** were normalized to ($-/-/-$) controls; ≥35 cells analysed per condition. **f** Changes to nuclear texture in immortalised hMSCs following SUN2 OE and CTS ($p = 0.01$ for CTS vs. unstrained), ≥17 cells analysed per condition. Box-whisker plot center lines show medians, bounds of box show quartiles, whiskers show data spread and outliers determined by the Tukey method; significance determined by Dunnett's multiple comparison tests. See Supplementary Table 1 for sample sizes

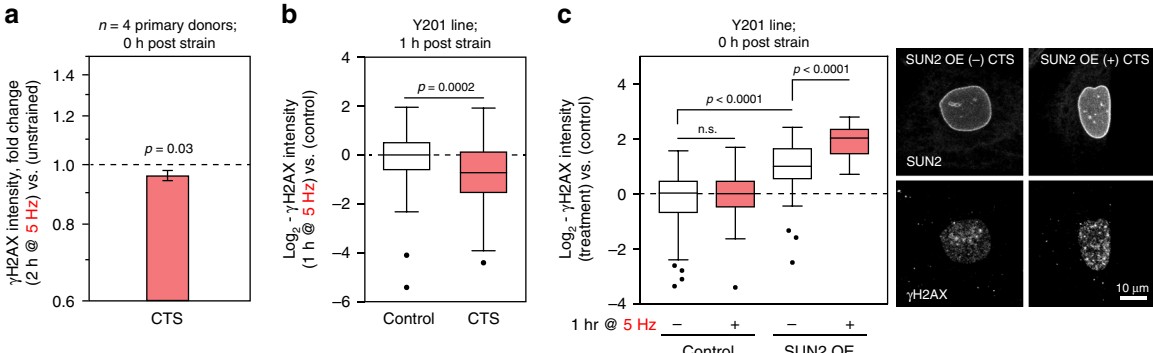

**Fig. 9** γH2AX staining is increased in SUN2 OE hMSCs subjected to CTS. **a** The integrated intensity of γH2AX stained foci was significantly decreased in primary hMSCs immediately following CTS ($p = 0.03$) (2 h, 2.6–6.2% strain at 5 Hz; $n = 4$ donors). See Supplementary Fig. 12a for data distributions, donor-to-donor variation and representative images. Plot shows mean ± s.e.m. $p$-values were determined from linear modeling (ANOVA). **b** γH2AX staining was also significantly decreased in immortalised hMSCs 1 h after CTS ($p = 0.0002$) (1 h, 2.6–6.2% strain at 5 Hz; ≥ 29 cells analysed per condition, $p$-value from two-tailed $t$-test). **c** Quantification of γH2AX staining in immortalised hMSCs with inducible SUN2 OE, immediately following CTS (1 h, 2.6–6.2% strain at 5 Hz). CTS significantly increased DNA damage in cells with SUN2 OE ($p < 0.0001$), ≥27 nuclei analysed per condition. $p$-values determined by Dunnett's multiple comparison tests. Box-whisker plot center lines show medians, bounds of box show quartiles, whiskers show data spread and outliers determined by the Tukey method. See Supplementary Table 1 for sample sizes

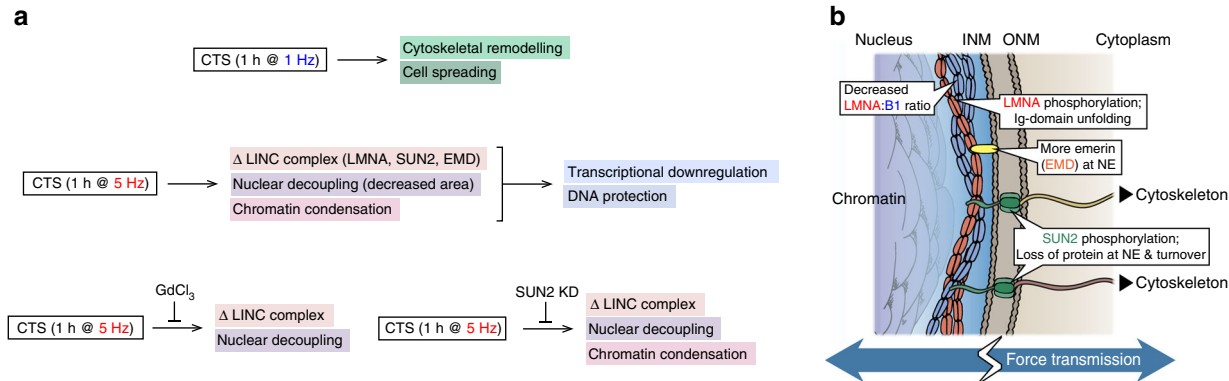

**Fig. 10** Effects of cyclic tensile strain (CTS) on mesenchymal stem cell (MSC) behavior. **a** Schematic summarizing the responses of MSCs to low and high intensity CTS, and how these pathways depend on ion channel activity and SUN domain-containing protein 2 (SUN2). **b** Cartoon summarizing strain-induced changes to proteins in the linker of nucleoskeleton and cytoskeleton (LINC) complex and nuclear envelope (NE). ONM outer nuclear membrane; INM inner nuclear membrane

multiple comparison tests; Fig. 9c). These results indicated that appropriate levels of SUN2 were essential for the mediation of nuclear decoupling in response to dynamic loading and therefore to afford protection to DNA.

## Discussion

We have demonstrated in primary cells from multiple donors that hMSCs have a rapid, structured and reversible response to CTS regulated at the protein level. This response was dependent on both functional ion channels and appropriate levels of the LINC complex protein SUN2 (Fig. 10a). Furthermore, CTS was shown to cause changes within the LINC complex (Fig. 10b), in particular to the regulation of SUN2, enabling cells to decouple nuclear and cellular morphological behaviors and conferring protection to DNA. Robustness is increased through cytoskeletal and nucleoskeletal remodeling in cells that have reached a mechanical equilibrium state on increasingly stiff substrates[56]. However, remodeling of the nuclear lamina seemed less important in the rapid response to high-intensity CTS. Mechano-transmission to the nucleus is an important mode of mechanical signaling, but if unregulated, has potential to apply stresses to chromatin. While a number of nuclear stress management mechanisms have been characterized, including chromatin condensation[29,34], chromatin detachment from the NE[57], and altered nuclear mechanics[1,7,58], a mechanism that isolates the nucleus from the cytoskeleton, as demonstrated here through regulation of SUN2, has potential to be both rapid and reversible. A role for SUN proteins in such mechanisms is further supported by analysis of protein turnover rates: SUN1 and 2 were reported to have the shortest half-lives of LINC complex proteins (Supplementary Fig. 12b)[59].

The -omics techniques described in this study have potential to explore broad aspects of the global cellular response to mechanical stress in greater detail. These include regulation of other structural proteins, such as the intermediate filaments[60], molecular chaperones and the pathways that manage DNA and oxidative damage. The use of MSCs as a model system to study mechano-responsive processes has been widespread, but these cells are also being assessed for their potential for therapy in heart[18] and muscle[17] – tissues subject to sustained and high-frequency deformation[61]. Furthermore, this work may be particularly relevant to understanding how mechanical stress contributes to age-related pathology. Many aspects of the cellular stress response are abrogated in ageing[62], but crucially, the NE may be particularly susceptible to misregulation[63,64].

## Methods

**Primary cell culture**. Human mesenchymal stem cells (hMSCs) were isolated from the bone marrow (knee and hip) of male and female donors using established methodology[65]. Informed written consent was obtained from donors. Experiments followed guidelines and regulations in accordance with the WMA Declaration of Helsinki and the UK Human Tissue Authority. All work was performed with approval from the NHS Health Research Authority National Research Ethics Service (approval number 10/H1013/27) and the University of Manchester). hMSCs were cultured on tissue culture treated polystyrene (TCTP) in low-glucose DMEM with pyruvate (Thermo Fisher Scientific) supplemented with 10% fetal bovine serum (FBS, Labtech.com) and 1% penicillin/streptomycin cocktail (PS, Sigma-Aldrich). For investigations into the effects of substrate stiffness, hMSCs were seeded onto type-I collagen coated polyacrylamide gels (2–50 kPa, Matrigen), and cultured in standard medium for three days.

**SUN domain-containing protein 2 (SUN2) protein knockdown**. hMSCs were incubated in complete medium containing RNAi Max Lipofectamine (Thermo Fisher Scientific) in the presence of short interfering double stranded RNA (siRNA; Thermo Fisher Scientific; final concentration of 10 nM):- SUN2 siRNA1 (s24467); sense: GGAAAUCCAGCAACAUGAAtt, antisense: UUCAUGUUGCUGGAU UUCCtc; SUN2 siRNA2 (s24467); sense: CCUUAGAGCAUGUGCCCAAtt, antisense: UUGGGCACAUGCUCUAAGGta.

A scrambled control was provided by the manufacturer (Thermo Fisher Scientific). Following 24 h of culture, the siRNA and Lipofectamine were removed. Cells were cultured for a further three days in complete media prior to further experimentation or analysis.

**Cyclic tensile strain (CTS)**. CTS was administered to cells using a FlexCell Tension Plus System (FX-4000T or FX-5000T; FlexCell International). Cells (primary hMSCs or transformed immortalised hMSCs, see following descriptions) were seeded onto type-I collagen coated BioFlex plates (FlexCell International) and cultured for 48 h to ensure adhesion. Cells were strained for 1 or 2 h at, low-intensity (0–4% strain at 1 Hz), intermediate-intensity (0–4% strain at 2 Hz), or high-intensity (2.6–6.2% strain at 5 Hz). Cells were fixed or lysed for downstream analysis either immediately following straining, or having been maintained in culture for a further 3 or 24 h.

**Microscopy and image analysis**. Cells were fixed with 4% paraformaldehyde (PFA, VWR International) in PBS for 10 min at RT, followed by 2 × 5 min washes in PBS. Cells were permeabilized using 1% Triton-X (Sigma-Aldrich) in PBS and blocked with 2% bovine serum albumin (BSA, Sigma-Aldrich), 0.25% Triton-X in PBS at RT for 30 mins. Cells were incubated overnight at 4 ℃ with primary antibodies against SUN1 (1:1000; Sigma, HPA008461), SUN2 (1:300; Sigma, HPA001209), LMNA/C (1:200; Santa Cruz Biotechnology, sc-7292), LMNB1 (1:2000; Abcam, ab16048), emerin (1:200; Leica Microsystems, NCL) and phospho-histone H2AX (S139) (γH2AX; 1:300; Merck, 05-636). Following 3 × 5 min PBS washes, cells were incubated with secondary antibodies, specific for mouse or rabbit IgG as appropriate, for 1 h at RT: AlexaFluor-488 goat anti-mouse (1:2000; ThermoFisher Scientific, A11029), AlexaFluor-594 donkey anti-rabbit (1:2000; ThermoFisher Scientific, A21207). Following further 3 × 5 min PBS washes, DAPI (1:1000; Sigma Aldrich, D9542) was used to stain cells at RT for 20 min; when used, AlexaFluor-488 Phalloidin (1:100; Cell Signaling Technology, #8878) was added with the DAPI stain. Samples were washed in PBS for 3 × 5 mins prior to imaging.

Images were captured on a Leica TCS SP5 or SP8 confocal microscope using HCX Apo U-V-I 20×/0.5 or HCX Apo U-V 63×/0.9 dipping lenses. Images were collected using hybrid detectors with the following detection mirror settings: green, 494–530 nm; red 602–665 nm; blue 420–470 nm. A white light source was filtered for excitation at 488 and 543 nm and a UV laser for excitation at 405 nm. The microscope used LAS X software (version 3.5, Leica); mages were processed in ImageJ (version 2.0.0, National Institutes of Health, USA); CellProfiler (version 2.1.1, Broad Institute, USA) was used to characterize cell morphometric parameters: cell area; cell perimeter; nuclear area; ratio of cytoplasmic to nuclear area. For quantification of proteins at the nuclear envelope (NE), the integrated fluorescence intensity was measured at the nuclear periphery, as identified by DAPI staining. To assess chromatin condensation, nuclei were imaged at 63x as described above and CellProfiler was used to quantify nuclear texture (Haralick texture features; sum variance scale 3; offset set to zero). For quantification of DNA damage, nuclei were imaged at ×63 and the integrated fluorescence intensity of nuclear phospho-histone γH2AX foci was quantified for each nucleus. Images were corrected for background fluorescence by subtracting the mean intensity of a cell-free area from each pixel; all images under comparison in the same experiment had matched exposure and contrast settings.

**Cell proliferation and viability assays**. Cell culture medium was removed from hMSCs immediately or 24 h after CTS (1 h at 1 or 5 Hz) and cells washed with PBS. Cells were lysed using 3× freeze/thaw cycles in PBS and lysates cleared using centrifugation at $1,2000 \times g$ for 10 min. Double-strand DNA concentration from each well was quantified using Quant-It PicoGreen Assay (Thermo Fisher Scientific), as described in the manufacturer's instructions. Fluorescence was recorded using a plate reader (excitation, 488 nm; emission, 520 nm). Concentrations were calculated from a standard curve generated with Lambda control DNA (Thermo Fisher Scientific).

Cell viability was measured in hMSCs immediately and 24 h after CTS using LIVE/DEAD Fixable Green Dead Cell Stain Kit (Thermo Fisher Scientific) in accordance with the manufacturer's instructions. Cells were washed in PBS and incubated with the viability dye diluted in PBS for 30 min at 37 °C. Cells were fixed using 4% PFA and imaged using a Leica TCS SP5 confocal microscope (×20 dipping lens). Cells killed with ethanol treatment were used as a positive control. The percentage of viable and dead cells was calculated from 6 random fields of view per treatment and per donor.

**RNA extraction**. RNA was extracted from cell pellets using the RNeasy Mini kit (Qiagen) as per the manufacturer's instructions. Briefly, cell pellets were thawed on ice and lysed using 350 μL of lysis buffer. In total 350 μL of 70% ethanol was added to each sample, the tubes mixed by inversion, and the solution drawn through the provided spin columns by centrifugation at $12,000 \times g$ for 30 s. The columns were washed with 350 μL of RW1 buffer using centrifugation ($12,000 \times g$ for 15 s) and an on-column DNA digest performed using the RNase-Free DNase kit (Qiagen), following the manufacturer's instructions. Briefly, 5 μL of DNase I enzyme was mixed with 35 μL of RDD buffer and added directly to the membrane of the spin columns. The columns were incubated at RT for 15 min. The columns were then washed with 350 μL of RW1 buffer using centrifugation ($12,000 \times g$ for 15 s), followed by an additional 2 × washes with 500 μL of RPE buffer and centrifugation. The RNA was eluted using 20 μL of water and the quality and quantity assessed using a NanoDrop ND-1000 spectrometer (Thermo Fisher).

**RT-qPCR**. In total 1 μg of RNA was reverse transcribed using the High Capacity RNA-to-cDNA Kit (ThermoFisher Scientific). RT-qPCR was performed in triplicate using SYBR Select Master Mix (ThermoFisher Scientific) using a StepOnePlus Real-Time PCR System (ThermoFisher Scientific). Data were analysed using the 2-ΔΔCt method[66] and normalized to *PPIA* and unstrained control cells. Custom designed and validated primers (PrimerDesign Ltd) were used as follows:- Vimentin (*VIM*); sense: TTCTCTGCCTCTTCCAAACTTT, anti-sense: CGTTGA TAACCTGTCCATCTCTA; Alpha-actin-2 (*ACTA2*); sense: AAGCACAGAGC AAAAGAGGAAT, anti-sense: ATGTCGTCCCAGTTGGTGAT; Peptidyl-prolyl isomerase A (*PPIA*); sense: ATGCTGGACCCAACACAAA, anti-sense: TTTC ACTTTGCCAAACACCA.

**RNA-Seq**. RNA-Seq analysis was performed by the Genomic Technologies Core Facility (GTCF) at the University of Manchester. In brief, strand-specific RNA-Seq libraries were prepared using the TruSeq Stranded mRNA Sample Preparation kit (Illumina). Data produced by an Illumina HiSeq4000 system was analysed with FastQC (Babraham Bioinformatics). In total 101 × 101 bp paired-end reads and between 24 and 124 M total reads were generated from each sample. Low quality reads and contaminated barcodes were trimmed with Trimmomatic[67]. All libraries were aligned to the *hg19* assembly of the human genome using TopHat (version 2.1.0; Center for Computational Biology, Johns Hopkins University) and only matches with the best score were reported for each read. The mapped reads were counted by genes with HTSeq[68] against gencode_v16.gtf. Log-transformed transcript fold changes were normalized under the assumption that the majority of genes were not perturbed by any of the experimental conditions.

**Protein labeling with monobromobimane (mBBr)**. Media was removed from cells immediately or 24 h after CTS treatment and cells were washed in PBS. Cells were then labeled by incubation with 2 mL of 400 μM monobromobimane (mBBr; Sigma-Aldrich) in PBS at 37 °C for 10 mins. Following labeling, 50 μL of 0.4 M glutathione in PBS was added to each well to quench the mBBr reaction. The quenched mBBr solution was removed and cells washed with PBS. Cells were detached from the substrate by incubating with 1 mL of trypsin at 37 °C for 10 min. Trypsin activity was neutralized using serum-containing culture medium and cells pelleted using centrifugation at $400 \times g$ for 5 min. Cells were resuspended in cold PBS, re-pelleted in 1.5 mL tubes (LoBind, Eppendorf) at $400 \times g$ for 5 min and cell pellets stored at −20 °C prior to proteomic analysis.

**Mass spectrometry (MS) sample preparation and analysis**. Six 1.6 mm steel beads (Next Advance) were added to the cell pellet tube with 30 μL SL-DOC (1.1% sodium dodecyl sulfate (Sigma), 0.3% sodium deoxycholate (Sigma), 25 mM ammonium bicarbonate (AB, Fluka), protease inhibitor cocktail (Sigma), sodium fluoride (Sigma), and sodium orthovanadate (Sigma) in de-ionized (DI) water). Cells were homogenized in a Bullet Blender (Next Advance) at maximum speed for 2 min. Homogenates were cleared by centrifugation (12 °C, $10,000 \times g$, 5 min).

Immobilized-trypsin beads (Perfinity Biosciences) were suspended in 150 μL of digest buffer (1.33 mM CaCl$_2$ (Sigma) in 25 mM AB) and 50 μL of protein lysate and shaken overnight at 37 °C. The resulting digest was then reduced (addition of 4 μL × 500 mM dithiothreitol (Sigma) in 25 mM AB; 10 min. shaking at 60 °C) and alkylated (addition of 12 μL × 500 mM iodoacetamide (Sigma) in 25 mM AB; 30 min. shaking at RT). Peptides were acidified by addition of 5 μL × 10% trifluoroacetic acid (Riedel-de Haën) in DI water, and cleaned by two-phase extraction (2 × addition of 200 μL ethyl acetate (Sigma) followed by vortexing and aspiration of the organic layer). Peptides were desalted, in accordance with the manufacturer's protocol, using POROS R3 beads (Thermo Fisher) and lyophilized. Peptide concentrations (measured by Direct Detect spectrophotometer, Millipore) in injection buffer (5% HPLC grade acetonitrile (ACN, Fisher Scientific) 0.1% trifluoroacetic acid in DI water) were adjusted to 300 ngμL$^{-1}$.

For hMSCs subjected to 5.0 Hz CTS, digested samples were analysed by LC-MS/ MS using an UltiMate 3000 Rapid Separation LC (RSLC; Dionex Corporation, Sunnyvale, CA) coupled to an Orbitrap Elite (Thermo Fisher Scientific, Waltham, MA) mass spectrometer. Peptide mixtures were separated using a gradient from 92% A (0.1% formic acid, FA (Sigma) in deionized water) and 8% B (0.1% FA in ACN) to 33% B, in 104 min at 300 nL per min, using a 75 mm × 250 μm inner diameter 1.7 μM CSH C18, analytical column (Waters). For analysis of SUN2 knockdown, digested samples were analysed by LC-MS/MS using an UltiMate 3000 RSLC (Dionex Corporation) coupled to a Q Exactive HF (Thermo Fisher Scientific) mass spectrometer. Peptide mixtures were separated using a multistep gradient from 95% A (0.1% FA in water) and 5% B (0.1% FA in ACN) to 7% B at 1 min, 18% B at 58 min, 27% B in 72 min and 60% B at 74 min at 300nLmin$^{-1}$, using a 75 mm × 250 μm inner diameter 1.7 μM CSH C18, analytical column (Waters). Peptides were selected for fragmentation automatically by data dependent analysis; mass spectrometers were operated using Xcalibur software (version 4.1.31.9, Thermo Scientific).

For assessment of SUN2 overexpression, identification of post-translational modifications (PTMs) in human MSCs following strain, and the response of MSCs to 1.0 Hz CTS, protein was extracted from cells by resuspension of cell pellets in 5% sodium dodecyl sulphate (SDS), 50 mM triethylammonium bicarbonate (TEAB), pH 7.55. Cell lysates (in 4 mm-thick microTUBEs, Covaris) were sonicated using a focused ultrasonicator (LE220-plus, Covaris) at 8 W for 21 min (sonicated for 300 s, peak power = 180, average power = 72, duty factor 40%, cycles per burst = 200, delay 15s, then repeated once). Samples were clarified using centrifugation at $13,000 \times g$ for 8 min. Samples were reduced by heating to 95 °C for 10 min in DTT at a final concentration of 20 mM. Cysteine was alkylated by addition of iodoacetamide to a final concentration of 40 mM and incubated at room temp in the dark for 30 min. Samples were cleared by centrifugation at $13,000 \times g$ for 8 min. Lysates were then acidified using aqueous phosphoric acid to a final concentration of 1.2% phosphoric acid and mixed with S-Trap binding buffer (90% aqueous methanol, 100 mM TEAB, pH 7.1). The protein lysate solutions were loaded onto S-Trap Micro Spin Columns by centrifugation at $4000 \times g$ for 1 min. The bound protein was washed three times using S-Trap binding buffer and then digested on column with trypsin (6 μg per sample) (Pierce, MS grade), reconstituted in digestion buffer (50 mM TEAB), for 1 h at 37 °C. Peptides were eluted in 50 mM TEAB, then 0.2% aqueous formic acid, and finally 50% acetonitrile containing 0.2% formic acid. Peptide concentration was quantified using a Direct Detect spectrophotometer (Millipore). Peptides were analysed using a Q Exactive HF (Thermo Fisher Scientific) mass spectrometer, as described above.

**Proteomics data processing**. MS spectra from multiple samples were aligned using Progenesis QI (version 4.1, Nonlinear Dynamics) and searched using Mascot (Matrix Science UK), against the SWISS-Prot and TREMBL human databases. Samples were not enriched for PTMs prior to MS (e.g. by affinity column), but MS spectra of samples from a SUN2 over-expressing immortalised cell line and primary MSCs subjected to CTS were aligned together to enable detection of PTMs to SUN2. The peptide database was modified to search for alkylated cysteine residues (monoisotopic mass change, 57.021 Da), oxidized methionine (15.995 Da),

hydroxylation of asparagine, aspartic acid, proline or lysine (15.995 Da) and phosphorylation of serine, tyrosine, threonine, histidine or aspartate (79.966 Da). In experiments in which cysteine residues were labeled with mBBr, the modification was searched for in two possible oxidation states (133.053 and 150.056 Da). A maximum of 2 missed cleavages was allowed. Peptide detection intensities were exported from Progenesis QI as Excel spreadsheets (Microsoft) for further processing.

Proteomics datasets were analysed using code written in-house in Matlab with the bioinformatics toolbox (R2015a, The MathWorks, USA). Raw ion intensities from peptides from proteins with fewer than 3 unique peptides per protein were excluded from quantification. Peptide lists were filtered leaving only those peptides with a Mascot score corresponding to a Benjamini-Hochberg false discovery rate (BH-FDR)[69] of <0.2. Normalization was performed as follows: raw peptide ion intensities were log-transformed to ensure a normal distribution and normalized within-sample by equalizing sample medians (subtracting sample median). Fold-change differences in the quantity of proteins detected in different samples were calculated by fitting a linear regression model that considers donor variability at both the peptide and protein levels. For each protein the following model was fit:

$$y_{fgd} = \beta_0 + X_f\beta_f + X_g\beta_g + X_d\beta_d + X_fX_g\beta_{fg} + X_fX_d\beta_{fd} + \varepsilon_{fgd} \quad (1)$$

Where $y_{fgd}$ represents the logged peptide intensity for peptide $f$, obtained under experimental treatment $g$, from donor $d$. $\beta$s represent the parameters to be estimated by the model fit, with $\beta_g$ and $\beta_{fg}$ representing the logged fold change in a given protein and peptide $f$, caused by experimental treatment $g$. Where peptide $f$ had been subject to post-translational modification (e.g. phosphorylation or reaction with mBBr), $\beta_{fg}$ would therefore give a measure of the fold-change in site occupancy (i.e. the fraction of the protein present that has been modified). $\beta_0$ is the intercept term that correlates with mean peptide intensity and $\varepsilon_{fgd}$ corresponds to a Gaussian error term centered on 0 with a width equal to the residual variance, $\sigma^2$. The model was fit using a Bayesian elastic net implemented using a Monte Carlo Markov Chain Gibb's sampler. Additionally, residuals were weighted according to their Mascot score each iteration of the Gibb's sampler such that outlier observations would not have their contribution to the model diminished if confidence in their identification was high. The full formulation and implementation is described in Mallikarjun et al.[36]. Standard error estimates were modified according to empirical Bayes correction of variances and empirical Bayes-modified $t$-tests were used to compare different experimental conditions. Linear modeling for Reactome pathway analysis was performed as described above using logged fold changes as the response variable according to the model for a given Reactome pathway:

$$y_{gp} = \beta_0 + X_g\beta_g + X_p\beta_p + \varepsilon_{gp} \quad (2)$$

Where $\beta_g$ and $\beta_p$ denote effect sizes due to experimental treatment $g$ and protein $p$. False positive correction of p-values for differential abundance was performed using Benjamini–Hochberg correction. Gene Ontology (GO) analysis was performed using Gorilla[32]. Resulting GO term lists were cleaned of redundant terms using REVIGO[33]. For cluster analysis, pathway enrichment was detected using Reactome[35], showing only terms with a false discovery rate (FDR) <0.05 and ≥3 entities associated. For analysis of phospho-proteomics, highlighted sites are those significantly differentially regulated (Benjamini–Hochberg FDR < 0.05) and observed previously by comparison to the PhosphoSitePlus database[70].

**Small molecule inhibition of ion channel activity**. hMSCs were cultured in complete medium containing inhibitors against stretch-activated ion channels 10 min prior to, and throughout, strain treatment, with vehicle only used as a control. Gadolinium chloride (GdCl₃; Sigma), dissolved in water and used at a final concentration of 10 μM (determined from a dose response experiment assessing its effects on nuclear morphology), was used as a broad-spectrum inhibitor against all stretch-activated ion channels[27]. Amiloride (Sigma), dissolved in DMSO and used at a final concentration of 100 μM (determined from a dose response experiment assessing its effects on nuclear morphology), was used as an inhibitor against acid sensing ion channels (ASICs)[27]. RN9893 (Sigma), dissolved in DMSO and used at a final concentration of 10 μM[28], was used to selectively inhibit the function of transient receptor potential vanilloid type 4 (TRPV4) ion channel. The tarantula venom peptide GsMTx4 (Abcam), dissolved in water and used at a final concentration of 3 μM[29] was used to inhibit piezo channels.

**Overexpression of SUN2 in an immortalised hMSC line**. The open reading frame of human SUN2 (isoform 2) was cloned into pCDH_TetOn, which contains a Tet response element enabling doxycycline-controlled expression of SUN2 protein. SUN2 (open reading frame); sense: AGACTCATCGCCACATTTCCA, antisense; AATCACACCTTCTTTCTGCAG; SUN2 (restriction enzyme sites added); sense (PacI): AAAAATTAATTAAATGTCCCGAAGAA, GCCAGCGC CTCACGCGCTAC, antisense (NheI): TTTTTGCTAGCCTAGTGGGCGG, GCT CCCCATGCACTCTGA.

Lentivirus was made by transfecting HEK 293 T cells in a T75 flask, with a complex of 4.5 μg of PsPax2 packaging vector, 3 μg of pMD2G packaging vector and 6 μg of pCDH_TetOn_Sun2, combined with polyethylenimine (PEI) (1 μgμL⁻¹) at a ratio of 1:2 DNA:PEI in DMEM medium and incubated at 37 °C overnight. The

transfection media was removed and replaced with complete medium containing sodium butyrate 10 mM (HDAC inhibitor) for 8 h, followed by replacement with complete medium. Following an additional 24 h culture in complete medium the virus containing media was removed, filtered through a Millipore Membrane 0.45 μm syringe filter (Merck) and stored at 4 °C, with fresh complete medium added to the HEK 293 T cells. Following an additional 24 h culture period, the second harvest of virus containing media was collected, filtered through a Millipore Membrane 0.45 μm syringe filter, combined with the first viral harvest and concentrated using a Vivaspin 20, 100 kDa MWCO PES column (Sartorius). Concentrated virus was aliquoted into 4× vials and stored at −80 °C until required.

The virus was used to transform immortalised hMSCs (line Y201[49]). The virus was added to 2 mL of complete medium containing polybrene 8 μgmL⁻¹ (Sigma) and added to 40% confluent Y201 hMSCs seeded in a 10 cm² dish. After 48 h of culture the virus containing media was removed and cells washed in complete medium twice. Cells were grown in culture for two weeks to allow transient viral expression to subside, then FACS sorted for blue fluorescent protein positive cells containing stable viral insertions.

Cells containing the inducible SUN2 expression vector were cultured for 4 days in complete medium containing 50 ngmL⁻¹ doxycycline (DOX, Sigma). Following DOX treatment, cells were cultured on either standard polystyrene tissue culture plastic, or type-I collagen coated BioFlex plates, prior to treatments and morphometric characterizations as described above. The length of time cells were cultured following DOX treatment was used to modulate levels of SUN2 expression. Non-transformed cells treated with 50 ngmL⁻¹ DOX were used as controls. For SUN2 rescue experiments, cells containing the inducible SUN2 expression vector were first treated with siRNA against SUN2 (siRNA2) as described above, with DOX then added for a further 4 days of culture. SUN2 rescue cells were then cultured on either standard polystyrene tissue culture plastic, or type-I collagen coated BioFlex plates, and morphometric analysis conducted.

**Chromatin condensation assay**. Primary hMSCs were cultured on type-I collagen coated BioFlex plates. They were treated for 30 min with complete medium containing MgCl₂ and CaCl₂ (both Sigma) at 2 mM final concentrations. Cells were fixed in 4% PFA, stained with DAPI (Sigma) for 20 min and imaged as described above.

**Statistical treatments and linear modeling**. Statistical tests were performed in Mathematica (version 11, Wolfram Research), MatLab (version R2015a, Math-Works) and GraphPad (version 8, Prism). All tests were two-tailed. Evaluations of R-squared and graphical analyses were performed using Igor Pro (version 6.37, Wavemetrics). Linear regression analysis was performed on imaging data where indicated, using the formula:

$$y_{gd} = \beta_0 + X_g\beta_g + X_d\beta_d + \varepsilon_{gd} \quad (3)$$

Where $y_{gd}$ represents a vector of normalized intensity data for each cell from experimental treatment $g$ and donor $d$. $\beta$s represent fold changes due to experimental treatment group $g$ and donor $d$ to be estimated by the model fit. $\beta_0$ represents the intercept term determined by the mean of $y$. Linear modeling was performed in Matlab using the *fitlm* function.

**Reporting Summary**. Further information on research design is available in the Nature Research Reporting Summary linked to this article.

## Data availability
Proteomics data have been deposited to the ProteomeXchange Consortium via the PRIDE partner repository with the identifiers: PXD012863, PXD012873, PXD012948, PXD012949 and PXD013287. RNA-Seq data is available via EMBL-EBI ArrayExpress with identifier E-MTAB-7925.

## Code availability
The BayesENproteomics code used to process MS data[36] is available to download from GitHub [https://www.github.com/VenkMallikarjun/BayesENproteomics].

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

## Acknowledgements

H.T.J.G. and J.S. were funded by a Biotechnology and Biological Sciences Research Council (BBSRC) David Phillips Fellowship (BB/L024551/1). V.M. was supported by a studentship from the Sir Richard Stapley Educational Trust. O.D. was supported by a Wellcome Institutional Strategic Support Fund (097820/Z/11/B). Proteomics was carried out at the Wellcome Centre for Cell-Matrix Research (WCCMR; 203128/Z/16/Z) Biological Mass Spectrometry Core Facility; RNA-Seq was performed by the Genomic Technologies Core Facility (GTCF); imaging was carried out at the Bioimaging Facility (supported by BBSRC, Wellcome and the University of Manchester Strategic Fund). We thank Professor Paul Genever (University of York, UK) for use of the Y201 immortalised hMSC line; Professor Tim Hardingham (University of Manchester, UK) for useful discussions; Drs. Ronan O'Cualain, Stacey Warwood, David Knight (MS), Ping Wang, Andrew Hayes (RNA-Seq), Craig Lawless, Julian Selley (bioinformatics), Steven Marsden, Roger Meadows and Peter March (bioimaging) for Core Facility support.

## Author contributions

Investigation, H.T.J.G., V.M., O.D., R.P., M.R.J. and J.S.; Formal Analysis, H.T.J.G., V.M., and J.S.; Writing – Original Draft, H.T.J.G.; Writing – Visualization, Review & Editing, H.T.J.G., V.M., O.D., M.R.J., R.P., A.P.G., SM.R. and J.S.; Project Administration and Funding Acquisition, J.S.

## Additional information

**Competing interests:** The authors declare no competing interests.

