## [Peer Review File · Nature Communications]

Reviewers' comments:

Reviewer #1 (Remarks to the Author):

Gilbert et al use cyclic tensile straining (CTS) of mesenchymal stem cells in combination with imaging, transcriptomics and proteomics to characterize the response of cells to mechanical forces. They discover an interesting difference between two regimes of CTS: low intensity stretching (1-2 Hz) induced cellular changes reminiscent of alterations that are observed in cells probing stiff and soft substrates, while high intensity stretching (5 Hz) induced decoupling of the nucleus from the cytoskeleton, which the authors speculate might protect chromatin from the mechanical forces. Inhibitor studies were used to show a dependence of this nuclear decoupling program on ion channels. Using transcriptomics the authors went on to show that high intensity CTS induced a general downregulation of transcription, which they speculate might be due to chromatin condensation. Protein quantification using mass spec was used to show that in contrast the response to high intensity CTS is stronger on protein level and not correlated with the response on mRNA level. Labeling of solvent exposed cysteines as well as mass spec analysis of PTMs was then used to show that CTS induces changes in protein conformation and post-translational modifications. Two specific examples for the protein response to CTS are discussed in more detail. The ratio of LMNA and LMNB1 was found to be unaltered, while the overall abundance and nuclear lamina location of SUN2 was decreased upon high intensity CTS. Both overexpression and knock-down of Sun2 affected nuclear area and affected the response to CTS. Overall, this work shows several interesting and important observations, is well written and presents good quality data and analysis. The reviewer suggests that some conclusions are not entirely supported by experimental evidence and has a few general and specific questions/comments below that may be addressed to improve the manuscript.

General comments

- The first part about nuclear decoupling in high intensity strain is much more convincing than the second part about the specific function of Sun2 in this process
- Description and analysis of PTM/mBBR experiments can be improved – currently this data is very descriptive and does not support any hypothesis put forward in this manuscript. Can the authors find an example of a regulated site that has a functional implication in the described protein response in nuclear decoupling?
- Sun2 knock down and overexpression both caused nuclear areas to increase. This is difficult to understand and not sufficiently well resolved. In Figure 7 the authors show that nuclear to cytoplasm ratio (which is decreased upon high intensity CTS) is not altered in Sun2 silenced cells. However this could be due to the also described general effect of Sun2 silencing on nuclear area (independent of the decoupling response). The authors should also show effects on cell area and nuclear area separately (not as ratio) to address this point.
- Can the authors visualize the nuclear linkage to the cytoskeleton under the relevant conditions using confocal imaging to directly support their model of a decoupling at the altered LINK complex?

Specific Comments

Figure 3

- all data needs to be provided at least in form of supplementary data tables. Individual outliers in scatter plots should be labeled with gene names.
- volcano plots of transcriptome and proteome data could be shown indicating the most significantly regulated genes/proteins
- panel e: protein and mRNA are not correlated, however some individual genes do correlate well. What are these? Where are the LINK complex proteins and in particular Sun2?

Figure 4

- all data needs to be provided at least in form of supplementary data tables. Individual outliers in scatter plots should be labeled with gene names.

- methods lack detail on how site occupancy was calculated
- all PTMs including mBBR labeling were determined simultaneously without enrichment? (better description in methods/results) If so the data suggests that most changes are not correlated (e.g. most phosphosite changes do NOT alter protein abundance, etc) – the text reads different and thus the conclusions should be more careful here
- the authors should examine the sites with correlation to protein abundance for already known functions (e.g. phosphosite plus) or their structural implications and put their proteomic findings in a functional perspective. For instance an example of one of the mBBR labeled peptides could be followed. Can the authors show a gene category / PFAM domain / motif enrichment analysis for regulated PTMs/mBBR sites? What did these experiments reveal about LINC complex proteins and Sun2?

Reviewer #2 (Remarks to the Author):

In their manuscript, Gilbert et al. provide a powerful systems-level analysis of the time-dependent response of primary human mesenchymal stem cells (hMSCs) to cyclic tensile strain (CTS) using quantitative image analysis, proteomics, and transcriptomics. They provide evidence to suggest that at low levels of CTS (1 hour, 4% CTS at 1 Hz), hMSCs displayed morphological changes that matched those induced by increased substrate stiffness. However, when they increased the CTS from 1 to 5 Hz they found that hMSCs exhibited a decoupling of the coordinated behavior of cell and nuclear spreading observed in cells at equilibrium on stiffness-defined substrates. Interestingly, the authors go on to show that this CTS-induced nuclear contraction required stretch-activated ion channels using a panel of ion channel inhibitors. Moreover, they reported that hMSCs exposed to 1 hour of CTS at 5 Hz rapidly established a widespread and reversible protein-level response despite the down-regulation of transcription. Through a quantitative analysis of protein abundance as well as conformation changes and post-translational modification, Gilbert et al. further reveal a suite of interesting changes to proteins known to participate in the mechanical coupling of the nucleus and the cytoskeleton via the LINC complex. Specifically, they find that hMSCs exposed to CTS (1 hr at 5 Hz) decrease the ratio of Lamin A/C:Lamin B1 and increase the phosphorylation status of Lamin A/C. In addition, they reveal that the nuclear envelopes of these cells exhibit increased and decreased levels of Emerin and SUN2, respectively. Consistent with SUN2 acting as a “strain-induced breakpoint”, the authors find that the depletion of SUN2 prevents the LINC complex protein changes, nuclear decoupling, and chromatin condensation typically induced by 1 hour of CTS at 5 Hz. Overall, this study provides important insights into the molecular responses of hMSCs to mechanical stimuli, which are critical for future investigations into their potential as therapeutics for use in mechanically active tissues including heart and skeletal muscle. Nevertheless, there are several major and minor issues that must be addressed before this manuscript can be accepted for publication.

Major Issues:

- 1) A major criticism of the manuscript that significantly reduces my enthusiasm for its publication in its current form is that it mostly lists the molecular responses of hMSCs to CTS without providing a discussion of their implications or the potential mechanisms responsible for those responses. This is particularly relevant regarding SUN2’s ability to act as a “strain-induced breakpoint”. Were any CTS-induced post-translational modifications in SUN2 identified (i.e. ubiquitin or phosphorylation) that might suggest a potential mechanism?
- 2) While they are interesting, the stretch activated ion-channel inhibitor experiments feel unfinished and possibly unnecessary. The fact that GdCl₃, RN9893, and amiloride each inhibited nuclear contraction following strain, albeit to different levels, suggests that multiple types of ion-channel are required for CTS-induced nuclear contraction. It seems that the authors need to use RNAi to test the role of specific candidate ion-channels during this process.
- 3) It is unclear to me how the proteins identified as “Cluster 3” in Figure 3F display a “sustained decrease” in expression, as the Z-score returns to 0 at 24s.

4) The Reactome Pathway "Rho GTPases activate IQGAPs", which is one of two pathways significantly affected by both SUN2 siRNAs relative to controls, is very confusing to me. The only gene that actually encodes a Rho GTPase within the cluster of "Rho GTPases activate IQGAPs" genes in Figure 6B is Cdc42.

5) Were the authors able to rescue the nuclear decoupling inhibited by SUN2 depletion by re-expressing SUN2 in SUN2-depleted cells? Based on Figures 5D, 5F, and 5H, it would appear not to be the case. This raises several concerns.

Minor issues:

1) What are the N.A.'s and the level of correction of the objectives used in this study?

2) In the Materials and Methods section, the authors state that they cloned the "open reading frame of SUN2 into pCDH_TetOn". Was this mouse or human SUN2? Which isoform?

3) The first sentence in the abstract: "Our current understanding of cellular mechano-signaling is based on static models, which do not replicate the dynamics of living tissues" is a very strong statement. Consider tuning it down or define the dynamics that are not recapitulated in static models.

4) Why was CTS chosen as the mechanical stimulus to apply to hMSCs in this study? This choice needs to be better motivated.

5) The authors need to provide a description of what the "Reactome pathways" are and why they are important for understanding the results presented in Figure 3.

6) Do the authors see phosphorylated FAK in response to CTS?

7) LMNA is a gene name. Use A-type nuclear lamins or lamin A/C. That being said, is it lamin-A or lamin-C or both?

8) In Supplemental Figure 4C, it appears that SUN1 is displaced from the nuclear envelope in response to CTS. Is this a real result? If so, can the authors provide some quantification?

9) In the section "SUN2 modulates transmission of CTS to the nucleus and DNA damage", the authors make the statement "This suggested that there was a LINC complex composition determined by the cellular microenvironment". However, it is unclear how they can make this statement.

Additional grammatical suggestions are provided in the supplemental file (Gilbert et al. Nat Comm Reviews 07-20-18) attached.

Reviewer #3 (Remarks to the Author):

A great deal of potentially interesting data underlies the presentation of the work, but it is at the onset bedeviled by a question of the physiological relevance of the applied strain regimen and over-interpretation of broad data sets.

1. The major issue with the paper is the relevance of the findings. The summary's first line, that "our current understanding mechano-signalling is based on static models", is incorrect. It is well known that cells respond to dynamic mechanical signals which have been studied extensively across cells, animals and humans, and that this response is critical to development, adaptation and repair, including in MSCs. In that light, the choice of a strikingly hyper-physiologic strain regimen of 4%, 5 Hz to MSC must be carefully considered, and justified, in terms of the biological relevance. Considering the extreme input variables, some evidence that the injury response of the cells (nuclear contraction, decrease in transcription/translation) is not dose dependent would be helpful (e.g., would 50 cycles/10 seconds be any different than 1 h of input). Further, it would be critical for the authors to provide evidence as to whether the MSC perceive such an input past the initial strain cycles – is it even possible to stretch an MSC 4% x 300 times in a minute?

2. The paper claims 2 major findings (summary). First that cellular protein is decreased after application of the hyper-physiological strain regimen: this should not be surprising given that the

cell is damaged. Secondly the authors demonstrate that SUN2 protein, a component of LINC, decreases by half within one hour, purportedly to “decouple mechano-transmission” to protect chromatin. While this may be the case, at this point it remains speculative without providing a mechanical dose challenge.

3. Fig 2's study of ion channel blockers seems to be a separate study as it is not taken up with respect to endpoints other than nuclear area/ texture.

4. With regards to CTS induced decrease in the intracellular proteome: the authors should evaluate whether this recovers at 24 h along with the return of cell shape. Data for changes in total proteome should also be provided for 1 Hz for comparison. The speculation that the decrease in protein is due to turnover/degradation should be addressed experimentally, at least for key proteins.

5. SUN2 protein in particular should be studied again at 24 h. They speculate, but do not prove, SUN2 loss is due to degradation versus decreased transcription. They fail to address whether any of the LINC proteins, in particular SUN2, are modulated by mBBR labeling, to approach a potential mechanism of why (or if) SUN2 protein is particularly targeted for decrease.

6. The SUN2 over-expression data is perplexing: in the cell they have chosen to show, the nuclear morphology is very abnormal, but no comment made. Nor do the authors offer any insight into how both knockdown and over-expression of SUN2 leads to nuclear area expansion.

7. The decrease in transcription was judged to be unimportant, not only recovering quickly, but perhaps unrelated to changes in protein structure/level. Yet the idea of “chromatin condensation” is offered as a potential major endpoint regarding the need for SUN2 to protect the chromatin (p 11). At the very least they should answer whether over-expression of SUN2 prevents chromatin condensation after 5Hz strain challenge.

8. The endpoints of nuclear size (plus/minus GdCl₃), SUN2 level and chromatin condensation should be provided for at least a 1 Hz application, and potentially for multiple magnitudes and frequencies.

9. Figure 7c regarding SUN2 states that “siRNA treatment was also sufficient to block changes to nuclear texture indicative of chromatin condensation” – the graph shows that this may be true for one of the siRNAs, but not for the other.

10. Figure 7d: contraction of nucleus in the “control” immortalized cell line is very small (if at all), making it difficult to say that the over-expression prevents the nuclear response to CTS. SUN2 OE cell should have an enlarged nucleus, but does not appear to in the cell line, why is this?

RESPONSE TO REVIEWER COMMENTS

Nuclear decoupling is part of a rapid protein-level cellular response to high-intensity mechanical loading

Gilbert et al.

Editor's comments

“Your manuscript entitled "Nuclear decoupling is part of a rapid protein-level cellular response to high-intensity mechanical loading" has now been seen by 3 referees, whose comments are appended below. You will see from their comments copied below that while they find your work of considerable potential interest, they have raised quite substantial concerns that must be addressed.”

Response: We thank the reviewers and editorial team for their enthusiasm for the work and the constructive comments that they have provided on the manuscript. We believe that we have been able to address the reviewers' concerns, and that this has resulted in a substantially improved manuscript.

“Notably, we require additional evidence elucidating the role of Sun2, particularly any CTS-induced PTMs on Sun2 or other proteins (all three reviewers), and the concerns about Sun2 overexpression and knockdown data (reviewers #1 and #3). In addition, thoroughly addressing the criticisms of the study design and relevance (see reviewer #3 comments) is also required.

[We] would be interested in considering a revised version that addresses these serious concerns with new experiments. All other points raised by the referees regarding the strength of the data (controls, validation experiments, quantifications etc.) must also be addressed, with additional experimentation as appropriate.”

Response: We have undertaken extensive new experimentation to enable us to address the concerns of the reviewers. Specifically, we have (i) used mass spectrometry to identify CTS-induced PTMs on SUN2; (ii) examined short-duration CTS to establish the sequence of decoupling phenomena; (iii) examined the effects of 1 Hz CTS on the proteome; (iv) examined the effects of SUN2 OE on cellular remodelling; and, (v) added 'rescue' controls to KD and OE experiments. New experiments are accompanied by updates to the text addressing experimental design, interpretation and relevance.

“If the revision process takes significantly longer than three months, we will be happy to reconsider your paper at a later date, as long as nothing similar has been accepted for publication at Nature Communications or published elsewhere in the meantime.”

Response: We thank the editor for granting us an extension to the resubmission deadline that has enabled us to conduct significant new experimentation.

“When resubmitting your paper, please highlight all changes in the manuscript text file.”

Response: We have updated the text, with changes indicated by **red type** in the accompanying document. There follows a point-by-point breakdown of our response to the reviewers’ comments.

“To improve the quality of methods and statistics reporting in our papers, we are now asking all authors to complete an editorial policy checklist that verifies compliance with all required editorial policies.”

Response: Please find the completed “Editorial Policy Checklist” form accompanying this document and a table specifying where accompanying data is available from.

Reviewer 1

“The first part about nuclear decoupling in high intensity strain is much more convincing than the second part about the specific function of Sun2 in this process.”

Response: We are pleased that the reviewer found the report of nuclear decoupling relatively convincing. We believe that this revision, which incorporates substantial new experimentation on SUN2 function (e.g. identification of PTMs, proteomic consequences of SUN2 overexpression, and restored phenotype with a ‘rescue’ of SUN2 levels), has helped to consolidate the second half of the manuscript.

“Description and analysis of PTM/mBBR experiments can be improved – currently this data is very descriptive and does not support any hypothesis put forward in this manuscript. Can the authors find an example of a regulated site that has a functional implication in the described protein response in nuclear decoupling?”

Response: The results regarding mBBR labelling, phosphorylation and oxidation (**Fig. 4** of the original manuscript, **Fig. 3** in the revision) give an untargeted overview of the cellular response to CTS. We believe this novel experiment to be highly informative, suggesting global changes in protein conformation, phosphoregulation and oxidative damage, all of which were reversible within 24 hours. Nonetheless, we agree that we had overinterpreted the evidence for correlation between protein conformation change and turnover. This statement has been revised as follows:

“This analysis showed that despite the suggestion of a link between CTS-induced changes to protein conformation and stability (rates of translation vs. turnover) on average (Figs. 2c, 3b), correlation on a protein-by-protein basis was low (Fig. 3c).”

The reviewer is correct in pointing out that a detailed interpretation of the role of an individual modification is very difficult without additional characterisation. We do this for two proteins: lamin-A/C because it has a well characterised, mechano-sensitive conformation (of the Ig-domain) and phosphorylation state (described in Swift *et al. Science* **2013**, a comparison between this work and ours is presented in **Supplementary Fig. 4g**; Buxboim *et al. Curr. Biol.* **2014**); and SUN2, where we identify mechano-sensitive regulation of location, abundance and post-translational modification (**Fig. 4**). Both of these proteins have direct relevance to the LINC complex and the ‘nuclear decoupling’ phenomenon.

“Sun2 knock down and overexpression both caused nuclear areas to increase. This is difficult to understand and not sufficiently well resolved.”

Response: We have examined this phenomenon in greater detail, performing new SUN2 KD, OE and rescue experiments in the immortalised Y201 hMSC line. We now present analysis of nuclear area, cell area and the ratio between the two (**Figs. 5g-j**). The explanation provided in the text follows:

“To investigate the role of SUN2 in mechano-transmission to the nucleus, we examined the relationship between SUN2 levels and cellular morphology. Y201 cells were cultured on plastic for three days following KD, OE and rescue of SUN2 levels, and compared to controls (Fig. 5g). SUN2 KD was also examined in primary hMSCs (Supplementary Fig. 5e) and modulation of SUN2 level at the NE was confirmed in all cases by IF (Supplementary Fig. 5f). Previous reports have linked SUN2 OE with abnormally shaped nuclei⁵⁵ and we found a significant reduction in nuclear form factor in Y201 cells subjected to SUN2 KD (Supplemental Fig. 5g). Consistent with the reduction in nuclear size following SUN2 depletion after CTS at 5 Hz, we found a weak positive scaling relationship between SUN2 level and nuclear area in Y201 cells (Fig. 5h and Supplementary Fig. 5h). However, the effect of SUN2 level on cytoplasmic area was stronger, in keeping with our observations of SUN2-induced ‘inside out’ remodeling, and dominated the scaling of the nuclear to cytoplasmic ratio (Figs. 5i, j, Supplementary Figs. 5h, i). This contrast between the change in nuclear to cytoplasmic area ratio following CTS (Fig. 1g) versus remodeling following imposed modulation of SUN2 levels (Fig. 5j) perhaps reflects the difference in time scales over which these processes occur.”

In Figure 7 the authors show that nuclear to cytoplasm ratio (which is decreased upon high intensity CTS) is not altered in Sun2 silenced cells. However, this could be due to the also described general effect of Sun2 silencing on nuclear area (independent of the decoupling response). The authors should also show effects on cell area and nuclear area separately (not as ratio) to address this point.”

Response: Data that showed nuclear-to-cytoplasmic area ratios in **Fig. 7** of the original manuscript (now in **Fig. 6** of the revised version) are now accompanied by separate plots of nuclear and cytoplasmic areas (**Supplementary Figs. 6b-e, g, h**). Perhaps because of the influence of SUN2 level on cytoskeletal remodelling (see **Fig. 5** and associated discussion), we found the nuclear-to-cytoplasmic ratio to be the most robust indicator of the ‘decoupling’ phenomenon and have now used it consistently throughout the manuscript (e.g. **Fig. 1g** has now been promoted to the main manuscript from the Supplementary Information).

“Can the authors visualize the nuclear linkage to the cytoskeleton under the relevant conditions using confocal imaging to directly support their model of a decoupling at the altered LINK complex?”

Response: This would be a great experiment, but is beyond our current capabilities. Our experimental system, the FlexCell5000, is robust and amenable to scale up. This has enabled us to perform -omics experiments (which require large numbers of cells), and high-content imaging (on fixed cells), using cells from multiple primary donors in parallel. In these respects, the system is a powerful tool enabling original aspects of study. However, the system is not compatible with live imaging during the application of strain, which would require complex control systems to keep individual cells within the field of view and focal plane during rapid deformation of the substrate. This is something we will seek to address in future studies.

“Figure 3

all data needs to be provided at least in form of supplementary data tables. Individual outliers in scatter plots should be labeled with gene names [...] volcano plots of transcriptome and proteome data could be shown indicating the most significantly regulated genes/proteins.

panel e: protein and mRNA are not correlated; however some individual genes do correlate well. What are these? Where are the LINK complex proteins and in particular Sun2?”

Response: [note that **Fig. 3** in the original manuscript is now **Fig. 2**] All proteomics data have been uploaded to the PRIDE data repository and are currently available to the public, along with associated metadata:

“Proteomics data have been deposited to the ProteomeXchange Consortium via the PRIDE partner repository⁶⁹ with the identifiers: PXD012863, PXD012873, PXD012948, PXD012949 and PXD013287.”

Nonetheless, we have also provided an EXCEL spreadsheet summarising transcriptomic and proteomic data used to make these plots. Outlying points, SUN2 and LMNA have now been annotated in all transcript/proteome and volcano plots (**Figs. 2e, Supplementary Figs. 2b, d, e; 3b, e**). On the last point, it would seem counterintuitive to use the limited space we have available to discuss correlating proteins/genes when our principle conclusions were that transcript was minimally perturbed and that correlation, if any, was very weak ($R\text{-squared} = 0.002$). Nonetheless, there may be interesting relationships in the data that we have not specifically identified, and other researchers would be able to find these in the datasets that we have made available online.

“Figure 4

all data needs to be provided at least in form of supplementary data tables. Individual outliers in scatter plots should be labeled with gene names.”

Response: [note that **Fig. 4** in the original manuscript is now **Fig. 3**] As mentioned in the previous response, all proteomics data have been made available via the PRIDE repository. Data from this plot is also included in the accompanying EXCEL spreadsheet. Individual outlier points in **Figs. 3c, e, g** are now labelled with the gene name and site of modification.

“methods lack detail on how site occupancy was calculated”

Response: An exhaustive description of the computational linear-modelling methods used to process the proteomics data, and specifically how they deal with donor-to-donor variability, post-translational modifications and pathway analysis can be found in Mallikarjun et al. (*BioRxiv*, 2018). Nonetheless, we have given a specific definition of how ‘site occupancy’ was evaluated in the Materials and Methods section of the Supplementary Information:

“Where peptide f had been subject to post-translational modification (e.g. phosphorylation or reaction with mBBR), $\beta_{f,g}$ would therefore give a measure of the fold-change in site occupancy (i.e. the fraction of the protein present that has been modified).”

The code for linear-modelling has also been made available to download via GitHub:

“The code used to process MS data⁴⁰ is available to download from:
www.github.com/VenkMallikarjun/BayesENproteomics”

“all PTMs including mBBR labeling were determined simultaneously without enrichment? (better description in methods/results) If so the data suggests that most changes are not correlated (e.g. most phosphosite changes do NOT alter protein abundance, etc.) – the text reads different and thus the conclusions should be more careful here”

Response: We did not enrich/prefractionate samples for modified peptides (such as enrichment of phosphopeptides by affinity column described, for example, in Olsen and Mann, *Mol. Cell. Proteomics*, **2013**). We have clarified this in the Materials and Methods section:

“Samples were not enriched for PTMs prior to MS (e.g. by affinity column), but MS spectra of samples from a SUN2 over-expressing immortalized cell line and primary MSCs subjected to CTS were aligned together to enable detection of PTMs to SUN2.”

The reviewer is correct that the plot of phosphorylation vs. protein abundance (**Fig. 3e**) does not suggest correlation and we have adjusted the wording of our interpretation accordingly. In addition to modifications noted above, we have edited the text to state:

“Phosphorylation of LMNA has been shown to be lower on stiffer substrates where total LMNA was increased^{1,7}. However, here we detected a modest (~1.1-fold) but significant increase in phosphorylation at S22, S390, S392, and S636, and no change in LMNA abundance. A plot of all changes in phosphosite occupancy vs. changes in abundance of the phosphorylated protein (Fig. 3e) did not exhibit general correlation, indicating that although phosphorylation may in many cases be mechanosensitive, it does not necessarily regulate turnover.”

“the authors should examine the sites with correlation to protein abundance for already known functions (e.g. phosphosite plus) or their structural implications and put their proteomic findings in a functional perspective. For instance, an example of one of the mBBR labeled peptides could be followed. Can the authors show a gene category / PFAM domain / motif enrichment analysis for regulated PTMs/mBBR sites? What did these experiments reveal about LINK complex proteins and Sun2?”

Response: LINC complex proteins did not show any changes in mBBR labelling. However, a more focused analysis of PTMs on SUN2 (previously reported in PhosphoSite and known to be in the lamin-binding domain) following CTS showed changes in phosphorylation which are detailed in **Fig. 4** and associated discussion:

“To better understand the regulation of SUN2 and its role in the nuclear decoupling phenomena, we examined the response of hMSCs to shorter durations of CTS at 5 Hz. IF showed that SUN2 was significantly reduced at the NE following 1 minute of CTS ($p = 0.002$; Fig. 4c). This preceded changes to the ratio of nuclear to cytoplasmic area, which were not significantly reduced within 10 minutes of CTS (Fig. 4c). PTMs have been shown to regulate the assembly of nuclear proteins such as LMNA^{1,7}, so we used MS to search for modifications to SUN2 following 1 hour of CTS at 5 Hz (coverage of the SUN2 amino acid sequence shown in Figs. 4e, f). This analysis uncovered four strain-responsive phosphorylation sites within the lamin-binding domain of SUN2; modifications were found that occurred immediately (pS12, pS21, and pS38) and some persisted 24 hours following CTS (pT9, pS12, and pS21; Fig. 4g).”

“MS also revealed changes to the phosphorylation state of SUN2 following OE (Fig. 5f): consistent with OE driving removal of excess protein, we found the same sites to be affected as when SUN2 was lost following CTS (Fig. 4g).”

Reviewer 2

“1) A major criticism of the manuscript that significantly reduces my enthusiasm for its publication in its current form is that it mostly lists the molecular responses of hMSCs to CTS without providing a discussion of their implications or the potential mechanisms responsible for those responses. This is particularly relevant regarding SUN2’s ability to act as a “strain-induced breakpoint”. Were any CTS-induced post-translational modifications in SUN2 identified (i.e. ubiquitin or phosphorylation) that might suggest a potential mechanism?”

Response: We have performed substantial additional work to address this criticism. Firstly, we examined the response of SUN2 levels and cell morphology in hMSCs exposed to shorter periods of CTS at 5 Hz. This enabled us to establish that significant SUN2 depletion at the nuclear envelope occurred very rapidly (within one minute), preceding changes in the nuclear-to-cytoplasmic area ratio, that were not significant even at ten minutes of CTS (**Figs. 4c, d**).

We then used mass spectrometry to search for post-translational modifications to SUN2 protein following CTS. By ‘aligning’ spectra with a sample containing SUN2 overexpression, we were able to increase our coverage of the SUN2 amino acid sequence and detect a number of phosphorylation sites (previously catalogued in the PhosphoSite database; **Figs. 4e, f**). We found that four of the phosphorylation sites in the lamin-binding domain of SUN2 were mechanosensitive (**Fig 4g**). We found the same sites to be affected when SUN2 was overexpressed (**Fig. 5f**). These observations enabled us to build a putative mechanism / sequence of events for the ‘nuclear decoupling’ phenomenon (**Fig. 4h**). This work is described in the section:

“The lamin-binding domain of SUN2 contains mechano-sensitive phosphosites

To understand the regulation of SUN2 and its role in the nuclear decoupling phenomena, we examined the response of hMSCs to shorter durations of CTS at 5 Hz. IF showed that SUN2 was significantly reduced at the NE after 1 minute of CTS ($p = 0.002$; Fig. 4c). This preceded changes to the ratio of nuclear to cytoplasmic area, which were not significantly reduced within 10 minutes of CTS (Fig. 4c). PTMs have been shown to regulate the assembly of nuclear proteins such as LMNA^{1, 7}, so we used MS to search for modifications to SUN2 following 1 hour of CTS at 5 Hz (coverage of the SUN2 amino acid sequence shown in Figs. 4e, f). This analysis uncovered four strain-responsive phosphorylation sites within the lamin-binding domain of SUN2; modifications were found that occurred immediately (pS12, pS21, and pS38) and some persisted 24 hours following CTS (pT9, pS12, and pS21; Fig. 4g).

Taken together, this evidence suggests a putative mechanism (Fig. 4h) whereby high-intensity CTS causes a rapid translocation of SUN2 from the NE – potentially mediated by phosphorylation of the lamin-binding domain – followed by a slower remodeling of cell and nuclear morphology and the cellular proteome (including turnover of SUN2) [...]

We accept that there is still work to be done to thoroughly explain the ‘nuclear decoupling’ behaviour, but we would suggest this to be beyond the scope of this (already extensive) study. There is some evidence in the paper that physical decoupling of the LINC complex (the “strain induced breakpoint”), SUN2 phosphorylation, SUN2 turnover, and cellular remodelling may not occur, or recover, at the same rate. This is the subject of ongoing investigation. We have not been able to determine whether phosphorylation of SUN2 precedes translocation from the nuclear envelope because mass spectrometry cannot give us sufficient time resolution (we don’t have means to ‘fix’ the cells, as in immunofluorescence). Development of phospho-specific antibodies, or loss/gain-of-function point mutations to SUN2 would provide additional insight, but this would be a considerable experimental undertaking. The loss/gain-of-function scheme is particularly difficult because (i) we don’t know what combinations of the four phosphosites are important; (ii) all current evidence suggests that controlling the level of SUN2 expression would be crucial. Anecdotally, the paper that identified mechano-sensitive phosphorylation of lamin-A/C (Swift et al. *Science* **2013**) required a year of further study before reports of loss/gain-of-function experiments were published in a subsequent paper (Buxboim et al. *Curr. Biol.* **2014**) – this was in a system where phospho-antibodies were already commercially available.

“2) While they are interesting, the stretch activated ion-channel inhibitor experiments feel unfinished and possibly unnecessary. The fact that GdCl₃, RN9893, and amiloride each inhibited nuclear contraction following strain, albeit to different levels, suggests that multiple types of ion-channel are required for CTS-induced nuclear contraction. It seems that the authors need to use RNAi to test the role of specific candidate ion-channels during this process.”

Response: We believe it was important to address ion channels to some extent in order to put this study in the context of seminal work by Wickström and Mauck groups (amongst others). However, we appreciate the comments of this reviewer and **Reviewer 3** and have accordingly reduced the amount of emphasis placed on this work and moved data associated with the former **Fig. 2** to the **Supplementary Information** section. We have not pursued any additional experiments.

“3) It is unclear to me how the proteins identified as “Cluster 3” in Figure 3F display a “sustained decrease” in expression, as the Z-score returns to 0 at 24h.”

Response: This is a good point. Cluster 3 is now described as “an immediate decrease and slow recovery”.

“4) The Reactome Pathway “Rho GTPases activate IQGAPs”, which is one of two pathways significantly affected by both SUN2 siRNAs relative to controls, is very confusing to me. The only gene that actually encodes a Rho GTPase within the cluster of “Rho GTPases activate IQGAPs” genes in Figure 6B is Cdc42.”

Response: The Reactome Pathway Database is a curated online resource (www.reactome.org) that attempts to classify genes/proteins by the pathways and functions with which they are reported to be involved (Croft *et al. Nucleic Acid Res.* **2014**; Fabregat *et al. Nucleic Acid Res.*, **2018**). It is a widely used bioinformatics tool (the 2014 paper has almost 900 citations). The annotation “Rho GTPases activate IQGAPs (R-HSA-5626467.1)” contains 32 proteins, including CDC42, but also RAC1, and proteins reported to be functionally associated with the pathway, such as ACTG1 and tubulins. Mass spectrometry proteomics is often unable to quantify small or low copy number proteins, so may not be able to report on all components within a given pathway. However, our analysis is able to identify where the pathway as a whole is significantly perturbed, based on what is observable by MS (Mallikarjun *et al. BioRxiv*, **2018**). We have clarified this in the caption of **Fig. 5b**:

“(b) Pathways identified in the Reactome database³⁹ as significantly affected by both SUN2 KDs, relative to scrambled controls, shown with fold changes to constitutive proteins.”

“5) Were the authors able to rescue the nuclear decoupling inhibited by SUN2 depletion by re-expressing SUN2 in SUN2-depleted cells? Based on Figures 5D, 5F, and 5H, it would appear not to be the case. This raises several concerns.”

Response: Yes, we were able to rescue the nuclear decoupling behaviour in the Y201 immortalised hMSC line when SUN2 depleted by siRNA treatment was restored by DOX-induced re-expression. This experiment is shown in **Fig 6e**.

“Rescue of SUN2 expression levels restored the capacity to decouple nuclei from the cytoskeleton following CTS (Figs. 6d, e), confirming the importance of correct SUN2 expression levels for this phenomenon to occur”

“Minor issues:

1) What are the N.A.’s and the level of correction of the objectives used in this study?”

Response: We have added the following details to the Materials and Methods section:

“Images were captured on a Leica TCS SP5 confocal microscope using HCX Apo U-V-I 20x/0.5 or HCX Apo U-V 63x/0.9 dipping lenses.”

“2) In the Materials and Methods section, the authors state that they cloned the “open reading frame of SUN2 into pCDH_TetOn”. Was this mouse or human SUN2? Which isoform?”

Response: We have added the following details to the Materials and Methods section:

“The open reading frame of human SUN2 (isoform 2) was cloned into pCDH_TetOn, which contains a Tet response element enabling doxycycline-controlled expression of SUN2 protein.”

“3) The first sentence in the abstract: “Our current understanding of cellular mechano-signaling is based on static models, which do not replicate the dynamics of living tissues” is a very strong statement. Consider tuning it down or define the dynamics that are not recapitulated in static models.”

Response: This is a reasonable point. We have softened the first line of the abstract:

“Studies of cellular mechano-signaling have often utilised static models that do not fully replicate the dynamics of living tissues.”

“4) Why was CTS chosen as the mechanical stimulus to apply to hMSCs in this study? This choice needs to be better motivated.”

Response: We agree that the choice of mechanical perturbation is very important, and acknowledge that the regime chosen (4% at 5Hz for 1 hour) is an intensive mechanical input. However, it is known that certain tissues in the body can experience very high frequencies (e.g. muscle tissue), with values as high as 40 Hz recorded during muscle tremor and vibration (McAuley *et al. Exp. Brain Res.* **1997**). As such a frequency of 5 Hz is within physiological values. Furthermore, while the frequency of the human heartbeat is below 5 Hz, the magnitude of the strain is much greater than we apply here (Aletras *et al. J. Magn. Reson.* **1999**). We don’t have space to expand greatly on our motivation, but have included the following extra line:

“To explore mechanisms that allow cells to endure more challenging mechanical environments (e.g. the mechanical environments encountered within dynamic tensile tissues, including muscle, skin and cartilage), we increased the frequency of CTS to 5 Hz (change in strain = 3.6%).”

As this reviewer and others have correctly pointed out, studying the cellular response to active strain cycle is by no means unprecedented (and we have toned down our language accordingly), nonetheless, the decision to examine the response to mechanical input as a source of ‘stress’ (akin to a heat shock) distinguishes this work. Indeed, as we started the study it was our original hypothesis that we would see consequences on molecular chaperone regulation. This may be true (see **Fig. 2f**), but the untargeted proteomics suggested a more remarkable effect in regulation of the LINC complex.

“5) The authors need to provide a description of what the “Reactome pathways” are and why they are important for understanding the results presented in Figure 3.”

Response: Please see the response to point (4) on page 8. Although quite widely recognised, we hope that an unfamiliar reader will be able to understand the principles behind pathway analysis from the reference that we cite in the manuscript, “The Reactome pathway knowledgebase” (Fabregat *et al. Nucleic Acid Res*, **2018**).

“6) Do the authors see phosphorylated FAK in response to CTS?”

Response: Focal adhesion kinase (PTK2 or FAK) was not observed in the MS data sets for either 1 or 5 Hz CTS. Mass spectrometry proteomics is good at quantifying high abundance proteins, such as the structural proteins studied here, but struggles with low copy number proteins. It is also difficult to find specific sites of phosphorylation without an enrichment protocol, and this problem is again exacerbated when the extent of phosphorylation is low. We were able to examine modifications to SUN2 because we could align the mass spectra with a sample with SUN2 overexpression, making it easier to find the protein of interest along with any associated PTMs.

An antibody-based method would be better suited to looking for FAK phosphorylation, but it would be difficult to reconcile with the existing narrative of the paper, defeating the purpose of an untargeted proteomics approach and prompting the criticism that we haven’t examined a whole canon of other already-recognised mechano-sensitive proteins.

“7) LMNA is a gene name. Use A-type nuclear lamins or lamin A/C. That being said, is it lamin-A or lamin-C or both?”

Response: We consistently use capitalised italics to refer to human genes and capitalised plain type to indicate their protein products. We have added the following explanation of nomenclature to the first place ‘LMNA’ is referred to in the text:

“Earlier applications of mBBR labeling have been used to identify force-dependent unfolding of domains in spectrin⁴⁴ and nuclear lamin-A/C (henceforth the abbreviation ‘LMNA’ will be used to refer to the total protein products of the *LMNA* gene, comprising of both lamin A and C spliceforms)¹.”

... although perhaps not universal, this system has been used in past literature on lamin (e.g. Swift *et al. Science* **2013**), on *Wikipedia* etc.

The antibody that we have used in this study (Santa Cruz sc-7292) binds both spliceforms of lamin-A/C, so we are effectively quantifying both in our immunofluorescence analysis. Likewise, the majority of the amino acid sequence (and therefore the peptides resulting from tryptic digest) are also common to both species, meaning that mass spectrometry effectively quantifies the average change (differences in the magnitude of changes to lamin-A vs. lamin-C would here be reflected by a greater estimate of standard error). The question as to whether *LMNA* splicing is mechano-sensitive is interesting (see discussion in Al-Saaidi & Bross, *Chromasoma*, **2014**) and past evidence suggests that perhaps it is (Swift *et al. Science* **2013**); more specialized mass spectrometry analysis or Western blotting could provide additional evidence, but we considered this to be beyond the scope of our present study.

“8) In Supplemental Figure 4C, it appears that SUN1 is displaced from the nuclear envelope in response to CTS. Is this a real result? If so, can the authors provide some quantification?”

Response: We struggled to get good quality images of SUN1 and they often exhibited high background/non-specific binding (we show a representative image; error bars in **Fig. 4b** are correspondingly large). This perhaps reflects a poor antibody, difficulties in imaging cells on PDMS substrates, and that SUN1 expression was very low. We consider SUN1 expression to be low in hMSCs compared to SUN2, based on the fact that it was not detectable by mass spectrometry despite being of a similar size and sequence to SUN2 (**Fig. 4a**). We were able to quantify changes to SUN1 at the nuclear envelope (using DAPI staining to locate the nuclear periphery), finding no significant changes in cells exposed to CTS (**Supplementary Fig. 4b**), but would not be confident in extending our analysis beyond this.

“9) In the section “SUN2 modulates transmission of CTS to the nucleus and DNA damage”, the authors make the statement “This suggested that there was a LINC complex composition determined by the cellular microenvironment”. However, it is unclear how they can make this statement.”

Response: We agree that this statement was poorly worded and it has been removed.

“Additional grammatical suggestions are provided in the supplemental file (Gilbert et al. Nat Comm. Reviews 07-20-18) attached.”

Response: We thank the reviewer for providing these suggestions. They were, for the most part, incorporated into the revised manuscript (although many will subsequently have been lost during changes made to accommodate new data). We appreciate that sometimes the style of writing became stilted, particularly as we were trying to keep the word count in check!

Reviewer 3

“1. The major issue with the paper is the relevance of the findings. The summary’s first line, that “our current understanding mechano-signalling is based on static models”, is incorrect. It is well known that cells respond to dynamic mechanical signals which have been studied extensively across cells, animals and humans, and that this response is critical to development, adaptation and repair, including in MSCs.”

Response: We agree with the second point, which was also made by **reviewer 2**. The previously hyperbolic first line of the Summary has been softened to:

“Studies of cellular mechano-signaling have often utilised static models that do not fully replicate the dynamics of living tissues.”

“[...] the choice of a strikingly hyper-physiologic strain regimen of 4%, 5 Hz to MSC must be carefully considered, and justified, in terms of the biological relevance. Considering the extreme input variables, some evidence that the injury response of the cells (nuclear contraction, decrease in transcription/translation) is not dose dependent would be helpful (e.g., would 50 cycles/10 seconds be any different than 1 h of input). Further, it would be critical for the authors to provide evidence as to whether the MSC perceive such an input past the initial strain cycles – is it even possible to stretch an MSC 4% x 300 times in a minute?”

Response: Please see the discussion of relevance in the earlier response to **reviewer 2** (point 4, page 9). As the reviewer suggests, we have now examined the response to shorter duration strain-cycle (see responses to following points and **Fig. 4**).

“2. The paper claims 2 major findings (summary). First that cellular protein is decreased after application of the hyper-physiological strain regimen: this should not be surprising given that the cell is damaged. Secondly the authors demonstrate that SUN2 protein, a component of LINC, decreases by half within one hour, purportedly to “decouple mechano-transmission” to protect chromatin. While this may be the case, at this point it remains speculative without providing a mechanical dose challenge.”

Response: We did not find any compelling evidence for persistent cellular damage. The cells did not appear to detach from the substrate or undergo apoptosis, viability was not affected (**Supplementary Figs. 1b, d, e**) and almost all characterisations showed a return towards the original behaviour at 24 hours. The cells appeared to be ‘managing’ the stressful condition, and the mechanisms of that ‘management’ are effectively what we have characterised. Please refer to the response to **reviewer 2** on page 7 for a description of the additional work we have performed in order to better characterise the mechanism of SUN2-mediated nuclear decoupling. We have explored various aspects of “mechanical dose challenge”, with additional characterisations of both frequency (**Supplementary Fig. 3**) and duration (**Fig. 4**) of CTS.

“Fig 2's study of ion channel blockers seems to be a separate study as it is not taken up with respect to endpoints other than nuclear area/ texture.”

Response: We thought it important to include a consideration of ion channel activity given its recognised importance within the field (e.g. work of Wickström and Mauck) and, indeed, it does have a demonstrable effect on the ‘nuclear decoupling’ phenomenon (see summary in **Fig. 7a**). Nonetheless, we recognise the point made by **reviewers 2** and **3**, and have moved the associated experimental details to the **Supplementary Information**.

“4. With regards to CTS induced decrease in the intracellular proteome: the authors should evaluate whether this recovers at 24 h along with the return of cell shape. Data for changes in total proteome should also be provided for 1 Hz for comparison. The speculation that the decrease in protein is due to turnover/degradation should be addressed experimentally, at least for key proteins.”

Response: Data showing the recovery of the cellular proteome after 24 hours is provided in **Supplementary Figs. 2c-e**; furthermore, the recovery of mBBr labeling, protein phosphorylation and oxidation after 24 hours is shown in **Figs. 3 b, d, f**.

We have conducted a new study of the effects of 1 hour of CTS at 1 Hz on the cellular proteome, which is shown in **Supplemental Figs. 3a-e**. The following discussion has been added:

“For comparison, we also examined proteomic changes in response to low-intensity CTS (1 hour at 1 Hz, change in strain = 4.0%). We found changes to 1 Hz CTS to be less pronounced than those induced at 5 Hz (Supplementary Figs. 3a, b; Gaussian width = 0.30), and although Reactome analysis showed similar pathways to be affected (compare Fig. 2d to Supplementary Fig. 3c), we noted that many of the significantly affected proteins were associated with the cytoskeleton. As was observed following 5 Hz CTS, the proteome recovered 24 hours following strain (Supplementary Figs. 3d, e; Gaussian width = 0.26), although some changes to cytoskeletal proteins persisted. Transcripts associated with the cytoskeleton were also affected: vimentin (*VIM*) and alpha-actin-2 (*ACTA2*) were significantly increased immediately following and 24 hours after CTS, respectively (Supplementary Figs. 3f, g). These results are consistent with previously observed changes to cell morphology, and earlier characterizations of cellular responses to strain^{26, 43} and substrate stiffness¹, which were proposed to increase nucleoskeletal and cytoskeletal robustness to stress.”

As described in earlier responses (please refer to page 7), we have performed extensive additional experimentation to examine the regulation of SUN2.

“5. SUN2 protein in particular should be studied again at 24 h. They speculate, but do not prove, SUN2 loss is due to degradation versus decreased transcription. They fail to address whether any of the LINC proteins, in particular SUN2, are modulated by mBBr labeling, to approach a potential mechanism of why (or if) SUN2 protein is particularly targeted for decrease.”

Response: We apologise for not making the levels of SUN2 protein and transcript clear in the original manuscript. We have now emphasised this quantification: *SUN2* transcript was not significantly changed immediately or 24 hours after CTS at 5 Hz (caption of **Fig. 2a** and **Supplementary Fig. 2a**); SUN2 protein was significantly decreased immediately after CTS at 5 Hz (**Fig. 4a** and **Supplementary Fig. 2b**) and recovered after 24 hours (caption of **Fig. 4a** and **Supplementary Figs. 2c, d**); SUN2 protein was not significantly affected by CTS at 1 Hz, either immediately or 24 hours following strain (**Supplementary Fig. 3**).

We should clarify that cells were only subjected to mBBr labeling briefly following the CTS cycle. No LINC complex proteins were found to be labeled. Changes to SUN2 level and cellular morphology were also extensively characterised in the absence of mBBr, so we have no reason to believe that the treatment was responsible for modulating the behaviour of LINC proteins.

As we have described in response to earlier points (please refer to page 7), we have conducted additional experiments to investigate the mechanism of SUN2 regulation (**Fig. 4**).

“6. The SUN2 over-expression data is perplexing: in the cell they have chosen to show, the nuclear morphology is very abnormal, but no comment made. Nor do the authors offer any insight into how both knockdown and over-expression of SUN2 leads to nuclear area expansion.”

Response: We have added a comment on the abnormally shaped nuclei, citing earlier work (Donahue *et al. J. Virol.* 2016):

“Previous reports have linked SUN2 OE with abnormally shaped nuclei⁵⁵ and we found a significant reduction in nuclear form factor in Y201 cells subjected to SUN2 KD (Supplemental Fig. 5g).”

[As described in the response to **reviewer 1**]: We have examined the relationship between SUN2 and cell morphology in greater detail, performing new SUN2 KD, OE and rescue experiments in the immortalised Y201 hMSC line. We now present analysis of nuclear area, cell area and the ratio between the two (**Figs. 5g-j**):

“To investigate the role of SUN2 in mechano-transmission to the nucleus, we examined the relationship between SUN2 levels and cellular morphology. Y201 cells were cultured on plastic for three days following KD, OE and rescue of SUN2 levels, and compared to controls (Fig. 5g). SUN2 KD was also examined in primary hMSCs (Supplementary Fig. 5e) and modulation of SUN2 level at the NE was confirmed in all cases by IF (Supplementary Fig. 5f). Previous reports have linked SUN2 OE with abnormally shaped nuclei⁵⁵ and we found a significant reduction in nuclear form factor in Y201 cells subjected to SUN2 KD (Supplemental Fig. 5g). Consistent with the reduction in nuclear size following SUN2 depletion after CTS at 5 Hz, we found a weak positive scaling relationship between SUN2 level and nuclear area in Y201 cells (Fig. 5h and Supplementary Fig. 5h). However, the effect of SUN2 level on cytoplasmic area was stronger, in keeping with our observations of SUN2-induced ‘inside out’ remodeling, and dominated the scaling of the nuclear to cytoplasmic ratio (Figs. 5i, j, Supplementary Figs. 5h, i). This contrast between the change in nuclear to cytoplasmic area ratio following CTS (Fig. 1g) versus remodeling following imposed modulation of SUN2 levels (Fig. 5j) perhaps reflects the difference in time scales over which these processes occur.”

“7. The decrease in transcription was judged to be unimportant, not only recovering quickly, but perhaps unrelated to changes in protein structure/level. Yet the idea of “chromatin condensation” is offered as a potential major endpoint regarding the need for SUN2 to protect the chromatin (p 11). At the very least they should answer whether over-expression of SUN2 prevents chromatin condensation after 5Hz strain challenge.”

Response: We have performed the experiment that the reviewer suggests, finding that SUN2 overexpression in the Y201 cell line prevents chromatin condensation in response to CTS at 5 Hz, as indicated by a lack increase to the nuclear texture parameter (**Fig. 6f**).

“SUN2-depletion significantly decreased strain-induced changes to nuclear texture, indicating reduced chromatin condensation (Fig. 6c, Supplementary Fig. 6f). Likewise, SUN2 OE in immortalised hMSCs blocked the changes to the nuclear to cytoplasmic area ratio observed in controls cells following 1 hour of CTS at 5 Hz (with recovery after 24 hours, Figs. 6d, e, Supplementary Fig. 6g-i). Rescue of SUN2 expression levels restored the capacity to decouple nuclei from the cytoskeleton following CTS (Figs. 6d, e), confirming the importance of correct SUN2 expression levels for this phenomenon to occur. SUN2 OE was also found to prevent the increase to nuclear texture, associated with chromatin condensation, that was caused by CTS (Fig. 6f).”

“8. The endpoints of nuclear size (plus/minus GdCl₃), SUN2 level and chromatin condensation should be provided for at least a 1 Hz application, and potentially for multiple magnitudes and frequencies.”

Response: The consequences of 1-hour CTS treatment at 1, 2 and 5 Hz on cell morphology (i.e. nuclear and cytoplasmic areas) are shown in **Fig. 1**. The levels of SUN2 protein were quantified by mass spectrometry following CTS at 1 and 5 Hz (please see response to point (5) on page 13). Changes to nuclear texture (as a reporter of chromatin condensation) were quantified following CTS at 1 Hz, 5 Hz and 5 Hz + GdCl₃, and can be found in **Supplementary Fig. 1h**.

“9. Figure 7c regarding SUN2 states that “siRNA treatment was also sufficient to block changes to nuclear texture indicative of chromatin condensation” – the graph shows that this may be true for one of the siRNAs, but not for the other.”

Response: [now **Fig. 6c**] We have adjusted the annotation of this figure to show where significant differences do occur. We have corrected reference to the significance in the text and caption:

“SUN2-depletion significantly decreased strain-induced changes to nuclear texture, indicating reduced chromatin condensation (Fig. 6c, Supplementary Fig. 6f).”

“(c) Changes to nuclear texture in primary hMSCs following SUN2 KD and CTS. SUN2 KD significantly reduced strain-induced changes in nuclear texture.”

“10. Figure 7d: contraction of nucleus in the “control” immortalized cell line is very small (if at all), making it difficult to say that the over-expression prevents the nuclear response to CTS. SUN2 OE cell should have an enlarged nucleus, but does not appear to in the cell line, why is this?”

Response: Because of the influence of SUN2 level on cytoskeletal remodeling (shown in **Fig. 5**), particularly in KD and OE experiments where the cells have several days to adapt their proteomes, we found it difficult to characterise nuclear morphology outside the context of cellular spreading. Nuclei were indeed larger in the immortalised cells with more SUN2 (**Fig. 5h**), but the effect on cell spreading was much greater (**Fig. 5i**). We have therefore included corresponding data on nuclear spreading, cellular spreading and the nuclear-to-cytoplasmic area ratio (which we found to be the most robust reporter of ‘decoupling’, e.g. **Figs. 1h, 6e**), which can be found in either main figures or the Supplementary Information.

REVIEWERS' COMMENTS:

Reviewer #1 (Remarks to the Author):

The authors have performed substantial revisions to their manuscript. They have (i) used mass spectrometry to identify CTS-induced PTMs on SUN2; (ii) examined short-duration CTS to establish the sequence of decoupling phenomena; (iii) examined the effects of 1 Hz CTS on the proteome; (iv) examined the effects of SUN2 OE on cellular remodeling; and, (v) added 'rescue' controls to KD and OE experiments.

I am satisfied with the response to my concerns.

Reviewer #2 (Remarks to the Author):

Overall, I feel that the authors have thoroughly addressed the majority of the concerns that I voiced in my review of their original manuscript. Consequently, their revised manuscript is nicely improved and represents an important advance for the field of nuclear mechanotransduction. That being said, I still would like the authors to incorporate the following minor suggestions in their revision before publication.

1) I would strongly suggest that the authors change the neon green text in Figures 3C, 3E, 3G, and 5D to another color, as it is quite difficult to see clearly.

2) Could the authors provide information regarding the lasers as well as excitation and emission filters used to acquire the images presented in this manuscript? I apologize for not having asked for this information in my initial review of their original manuscript.

3) The authors should define the "SUN" acronym as "Sad1/UNC-84" when it is first used in their manuscript (i.e. the abstract), as opposed to page 9 where it is defined in their revised manuscript.

Reviewer #3 (Remarks to the Author):

The authors have addressed major issues by providing a great deal of new data and making substantial revisions to the figure set, thus enhancing their conclusions that SUN2 is altered in response to 5Hz strain. These revisions have significantly enhanced the overall work.

1. The addition of 1Hz in Supplemental fig3 is appreciated. The 'less pronounced' proteomic changes- (2C's 0.25 to 0.61 at 5 Hz compared to Supp3A's 0.25 to 0.3 at 1 Hz) are substantial. The authors should comment on the statistical significance of the 1Hz change - they suggest that the reactome pathways are significantly affected at $p < 0.05$ but is this biologically relevant? Are the same proteins affected as 5Hz?

2. New Fig 4 shows modifications to SUN2 after 1 h which is very interesting. The statement in p 10's text: "rapid translocation of SUN2 from the NE" is unclear. Fig 4H suggests "translocation of SUNS away from NE". Translocation may not be appropriate language.

Reviewer 1

The authors have performed substantial revisions to their manuscript. They have (i) used mass spectrometry to identify CTS-induced PTMs on SUN2; (ii) examined short-duration CTS to establish the sequence of decoupling phenomena; (iii) examined the effects of 1 Hz CTS on the proteome; (iv) examined the effects of SUN2 OE on cellular remodelling; and, (v) added 'rescue' controls to KD and OE experiments. I am satisfied with the response to my concerns.

Response: We are pleased that we have been able to address the reviewer's previous concerns.

Reviewer 2

Overall, I feel that the authors have thoroughly addressed the majority of the concerns that I voiced in my review of their original manuscript. Consequently, their revised manuscript is nicely improved and represents an important advance for the field of nuclear mechanotransduction. That being said, I still would like the authors to incorporate the following minor suggestions in their revision before publication.

Response: We thank the reviewer for the complimentary assessment of the revised manuscript.

1) I would strongly suggest that the authors change the neon green text in Figures 3C, 3E, 3G, and 5D to another colour, as it is quite difficult to see clearly.

Response: We agree that the neon green colour was difficult to read. We have now replaced all instances of neon green with a darker green that is more legible.

2) Could the authors provide information regarding the lasers as well as excitation and emission filters used to acquire the images presented in this manuscript? I apologize for not having asked for this information in my initial review of their original manuscript.

Response: The following has now been added to the Methods section: "Images were collected using hybrid detectors with the following detection mirror settings: green, 494-530 nm; red 602-665 nm; blue 420-470 nm. A white light source was filtered for excitation at 488 nm and 543 nm and a UV laser for excitation at 405 nm."

3) The authors should define the "SUN" acronym as "Sad1/UNC-84" when it is first used in their manuscript (i.e. the abstract), as opposed to page 9 where it is defined in their revised manuscript.

Response: "SUN" is now defined in the Abstract and upon its first use in the Introduction.

Reviewer 3

The authors have addressed major issues by providing a great deal of new data and making substantial revisions to the figure set, thus enhancing their conclusions that SUN2 is altered in response to 5Hz strain. These revisions have significantly enhanced the overall work.

Response: We are pleased that the reviewer appreciated the changes we have made to the manuscript.

1. The addition of 1Hz in Supplemental fig3 is appreciated. The 'less pronounced' proteomic changes—(2C's 0.25 to 0.61 at 5 Hz compared to Supp3A's 0.25 to 0.3 at 1 Hz) are substantial. The authors should comment of the statistical significance of the 1Hz change - they suggest that the Reactome pathways are significantly affected at $p < 0.05$ but is this biologically relevant? Are the same proteins affected as 5Hz?

Response: We are pleased that the reviewer found this additional analysis informative. The similarity in the Reactome analyses at 1 and 5 Hz perhaps masks some aspects of the data as the ontology classifications are quite broad. We have therefore considered the significance of changes to cytoskeletal proteins individually, and made a comparison between the 1 and 5 Hz experiments. Our biological interpretation of the response to 1 Hz strain is that the cells are remodelling their structural protein content to make them more robust (similar to the remodelled states on stiff vs. soft substrates). This perhaps contrasts some aspects of the response to 5 Hz strain, where cellular resources are directed toward protection against an acute stress (e.g. regulation of SUN2). The discussion of this experiment has been expanded correspondingly in the main text:

“For comparison, we also examined proteomic changes in response to low-intensity CTS (1 hour at 1 Hz, change in strain = 4.0%). We found changes to 1 Hz CTS to be less pronounced than those induced at 5 Hz (Gaussian width = 0.30; Supplementary Figs. 4a, b, Supplementary Data 8 and 9), and although Reactome analysis showed similar pathways to be affected (compare Fig. 2d to Supplementary Fig. 4c), we noted that many of the significantly affected proteins were associated with the cytoskeleton. CTS at 1 Hz caused a significant increase in levels of actin (ACTB), vimentin (VIM), tubulin alpha-1B chain (TUBA1B) (all FDR-corrected $p < 0.0001$), dynamin-2 (DNM2, $p = 0.04$) and the nucleoskeletal protein lamin-A/C (henceforth the abbreviation LMNA will be used to refer to the total protein products of the LMNA gene, comprising of both lamin A and C spliceforms; $p = 0.008$); myosin light chain 6B (MYL6B) and the mechanoresponsive transcriptional coactivator YAP1 were downregulated ($p = 0.001$ and 0.03 , respectively). Where these proteins were also detected in the 5 Hz experiment, only VIM was significantly affected (down-regulated, $p = 0.002$). As was observed following 5 Hz CTS, the proteome partially recovered 24 hours following strain at 1 Hz (Supplementary Figs. 4d, e; Gaussian width = 0.26), although some changes to cytoskeletal proteins persisted (TUBA1B remained elevated, ACTB was decreased; both FDR-corrected $p < 0.0001$). Transcripts associated with the cytoskeleton were also affected: vimentin (VIM) was increased immediately following CTS at 1 Hz ($p = 0.0003$) and alpha-actin-2 (ACTA2) was increased after 24 hours ($p = 0.004$) (Supplementary Figs. 4f, g). These results are consistent with previously observed changes to cell morphology, and earlier characterizations of cellular responses to strain^{23,39} and substrate stiffness¹, which were proposed to increase nucleoskeletal and cytoskeletal robustness to stress.”

2. New Fig 4 shows modifications to SUN2 after 1 h which is very interesting. The statement in p 10's text: "rapid translocation of SUN2 from the NE" is unclear. Fig 4H suggests "translocation of SUNS away from NE". Translocation may not be appropriate language.

Response: We agree that 'translocation' may not be the appropriate language as this may imply that proteins are reversibly transferred between two locations (e.g. mechanically induced translocation of YAP between nucleoplasm and cytoplasm). We don't currently have direct evidence of where SUN2 is moved to in response to mechanical stress (e.g. to Golgi, lysosome etc.) We have therefore substituted 'loss' in place of 'translocation' in both text and figures:

MAIN TEXT: "Taken together, this evidence suggests a putative mechanism (Fig. 5d) whereby high-intensity CTS causes rapid loss of SUN2 from the NE – potentially mediated by phosphorylation of the lamin-binding domain – followed by a slower remodeling of cell and nuclear morphology and the cellular proteome (including turnover of SUN2)."

FIGURE 4 CAPTION: "Significant loss of SUN2 at the NE occurred within 1 minute (1 minute, $p = 0.002$; 10 minutes, $p < 0.0001$). (d) Nuclear to cytoplasmic area ratios quantified following 1 and 10 mins of 5 Hz CTS. Red line indicates area ratio following 1 hour of CTS ($p = 0.003$; Fig. 1f). These results indicate that loss of SUN2 from the NE precedes changes to cellular morphology."

Further edit for clarification (not raised by reviewers):

We have edited the last paragraph of the Results section, adding reference to a p-value given in Fig. 9c:

"We were surprised, therefore, to find that CTS here resulted in a small but significant decrease in the intensity of γ H2AX staining in primary ($p = 0.03$) and immortalised MSCs ($p = 0.0002$), suggestive of a protective effect (Figs. 9a, b, Supplementary Fig. 7j). However, we found that the OE of SUN2 in immortalised hMSCs shown to override the decoupling response to CTS raised the baseline level of γ H2AX staining ($p < 0.0001$) and caused staining to be further increased immediately following CTS ($p < 0.0001$; Fig. 9c)."